# An engineered thermal-shift screen reveals specific lipid preferences of eukaryotic and prokaryotic membrane proteins

Emmanuel Nji [1], Yurie Chatzikyriakidou[1], Michael Landreh[2] & David Drew[1]

Membrane bilayers are made up of a myriad of different lipids that regulate the functional activity, stability, and oligomerization of many membrane proteins. Despite their importance, screening the structural and functional impact of lipid–protein interactions to identify specific lipid requirements remains a major challenge. Here, we use the FSEC-TS assay to show cardiolipin-dependent stabilization of the dimeric sodium/proton antiporter NhaA, demonstrating its ability to detect specific protein-lipid interactions. Based on the principle of FSEC-TS, we then engineer a simple thermal-shift assay (GFP-TS), which facilitates the high-throughput screening of lipid- and ligand- interactions with membrane proteins. By comparing the thermostability of medically relevant eukaryotic membrane proteins and a selection of bacterial counterparts, we reveal that eukaryotic proteins appear to have evolved to be more dependent to the presence of specific lipids.

[1] Centre for Biomembrane Research, Department of Biochemistry and Biophysics, Stockholm University, SE-106 91 Stockholm, Sweden. [2] SciLifeLab and Department of Microbiology, Tumor and Cell Biology, Karolinska Institutet, SE-171 65 Stockholm, Sweden. These authors contributed equally: Emmanuel Nji, Yurie Chatzikyriakidou. Correspondence and requests for materials should be addressed to D.D. (email: ddrew@dbb.su.se)

Membrane bilayers are assembled from a plethora of lipid classes with varying biophysical properties. These lipids are thought to regulate the functional activity, stability, and oligomerization state of many membrane proteins[1]. Despite this prevailing view, due to technical hurdles, interactions between lipids and membrane proteins are often overlooked and infrequently measured. Spectroscopy and computational prediction strategies have proven to be particularly powerful for analyzing individual lipid–protein interactions[2,3]. Recently, mass spectrometry (MS) has gained traction as a tool for unraveling the intricate connections between membrane proteins and lipids[4], shedding light on different lipid binding modes[5], lipid-mediated protein stabilization[6] and even thermodynamics and allosteric effects of protein–lipid interactions[7,8]. Although it has proven powerful for studying individual protein–lipid contacts and is progressing toward the analysis of complex mixtures[9], MS, like other spectroscopic and computational approaches, is to date not well-suited for identifying specific interactions in native membranes or in high-throughput formats, hampering any large-scale investigations.

Thermofluor-based thermal-shift assays are high-throughput strategies widely used for screening for the binding of small molecule libraries[10,11]. However, they cannot cope with unpurified samples, require large amounts of purified protein, and sometimes give uninterpretable results[12]. The fluorescence-detection size-exclusion chromatography-based thermostability assay (FSEC-TS) is an alternative method that does not require any endogenous ligands[13], but instead uses the intrinsic fluorescence of the green fluorescent protein (GFP)-fusion partner to monitor thermal denaturation. Encouragingly, it has been demonstrated that FSEC-TS is compatible with both unpurified and purified samples and could capture broadly stabilizing effects of a range of lipids on a pentameric chloride channel[13]. However, its ability to detect specific lipid–protein interactions has not been explored, and the requirement of size-exclusion step limits its applicability for large-scale investigations.

Here, we outline a GFP-based TS (GFP-TS), which is rapid and easy to perform, and enables quantification of lipid and ligand interactions with membrane proteins. Using this technology, we compare the sensitivity of bacterial and eukaryotic membrane proteins to the presence of various physiologically relevant lipids and identify selectively stabilizing interactions.

## Results

### Validating FSEC-TS for measuring lipid–protein interactions.

We have recently investigated lipid binding to sodium/proton antiporters using a combination of nondenaturing MS and molecular dynamics (MD) simulations. We could show that the sodium–proton antiporter NhaA from *E. coli*, but not the distantly related structural homolog NapA from *T. thermophilus*, is able to specifically retain a small amount of bound cardiolipin throughout purification[14]. MS and MD data suggest that in the lipid-bound fraction, the cardiolipin may act as a bidentate ligand that stabilizes the dimeric form of NhaA by bridging across subunits[15]. We, therefore, selected the interaction between cardiolipin and NhaA as a test-case for the detection of specific lipid binding by the FSEC-TS method. In brief, dodecyl-β-D-maltopyranoside (DDM) purified NhaA– and NapA–GFP fusions were heated over a range of different temperatures, aggregates sedimented by centrifugation, and the supernatant subjected to size-exclusion chromatography on an high performance liquid chromatography (HPLC)-system coupled with an in-line fluorescence detector. By measuring the decrease in GFP-fusion peak height at increasing temperatures, the apparent melting temperatures ($T_m$) for NhaA and NapA were calculated, Fig. 1a.

Consistent with the MS data[15], FSEC-TS strongly indicated cardiolipin (18:1) binding to NhaA as its apparent $T_m$ increased significantly from 42 to 53 °C. In contrast, NapA showed no improvement in stability, which indicated that it does not specifically bind cardiolipin. Because it has been shown that lipids can stabilize membrane proteins indirectly by increasing the associated detergent micelle size[16], we examined if any component of the stabilization effect was nonspecific. As such, we heated NhaA at 5 °C above its apparent $T_m$ and compared the thermostability of samples supplemented with either cardiolipin or phosphatidylglycerol (PG), which is highly abundant in *E. coli* cells. However, the addition of PG did not increase NhaA stability, Fig. 1b. Subsequently this assay was used to validate the role of cardiolipin in the dimerization of NhaA. For this purpose, we constructed and purified a monomeric variant of NhaA (NhaA-β-less), which lacks the β-hairpin extension required for dimerization[17,18] (see Methods). Strikingly, this monomeric mutant was not stabilized by the addition of cardiolipin (18:1), demonstrating that cardiolipin is indeed incorporated into the dimeric form of NhaA Fig. 1c.

In the dimeric NhaA crystal structure[17], additional nonprotein electron density was found between four interface arginine residues that, while modeled as DDM and two sulfate anions, could be reinterpreted as a binding site for cardiolipin, Fig. 1d. Consistently, crystal structures have shown that negatively charged cardiolipin has a preference to bind at positively charged surface clusters of arginine and lysine[19]. Having validated the use of the FSEC-TS method for detecting specific lipid interactions to membrane proteins, we further investigated if the proposed lipid-driven oligomerization could be quantified. To do so, we heated NhaA fusions at 47 °C (apparent $T_m + 5$ °C) at increasing the cardiolipin (18:1) concentrations. As shown in Fig. 1d, between a concentration of 1–3 mM cardiolipin (18:1) we indeed see a sharp increase in the stabilization of NhaA. From the FSEC traces it is apparent that the increase in stabilization of NhaA is directly correlated with an increase in the population of NhaA dimers, Fig. 1e. Thus, we conclude that the FSEC-TS method is suitable for detecting and quantifying lipid–protein interactions involving specific lipid species at specific binding sites.

### Developing GFP-TS for high-throughput lipid screening.

FSEC-TS is a relatively low to the mid-throughput method, as it requires a size-exclusion step for separating the protein peak from heat-induced aggregates. We, therefore, investigated whether the SEC-step could be replaced by centrifugation to accelerate and simplify the overall procedure, using the thermal stability of the bile acid sodium-coupled symporter homolog ASBT$_{NM}$, which belongs to the same fold as the sodium/proton antiporters[20], as an example. Since the GFP-fluorescence signal is detectable in crude material, we performed the thermal stability measurements on *E. coli* membrane fractions that had previously solubilized in 1% (w/v) DDM at final total protein concentration of 3.5 mg mL$^{-1}$ (see Methods).

Detergent-solubilized membrane samples containing recombinant ASBT$_{NM}$-GFP fusion were incubated at 4 °C–100 °C, aggregates sedimented by centrifugation, and the fluorescence in the supernatant measured using a 96-well plate spectrofluorometer, Fig. 2a. The same supernatant samples were individually analyzed by FSEC and their respective peak heights determined, Fig. 2b. However, without the size-exclusion step the calculated apparent $T_m$ was ~11 °C higher than that estimated by FSEC-TS. We reasoned that the poor correlation is due to insufficient removal of fluorescent aggregates by centrifugation alone, as was obvious from the FSEC-TS traces, Fig. 2b. In attempt to better clear the supernatant from fluorescent aggregates, we briefly

incubated the DDM-solubilized protein solutions with the short-chain nonionic detergent octyl-β-D-glucoside (β-OG) prior to heating, as we had previously observed that heat induced aggregates tend to precipitate in this detergent[21]. Indeed, the apparent $T_m$ determined after addition of β-OG and centrifugation closely matched that measured by the FSEC-TS method, Fig. 2c. Consistent with our hypothesis, there were no longer any fluorescent aggregates remaining in the FSEC-TS traces when β-OG was added to the DDM-solubilized membranes, Fig. 2d. To confirm our interpretation, we further measured the fluorescence

in the pellets formed after heating and centrifugation. Indeed, the decrease in supernatant ASBT_NM-GFP fluorescence at higher temperatures is now inversely correlated with an increase of pellet fluorescence up to 76 °C, Fig. 2a, c and Supplementary Fig. 1a; above this temperature GFP is itself no longer fluorescent[13].

To evaluate if we could reliably measure the thermostability of other membrane proteins under these conditions, we expanded our test-set to include the proteins AmtB (ammonium channel), EmrD (multidrug exporter), GlpG (rhomboid protease), GlpT (glycerol-3-phospahte transporter), Mhp1 (hydantoin

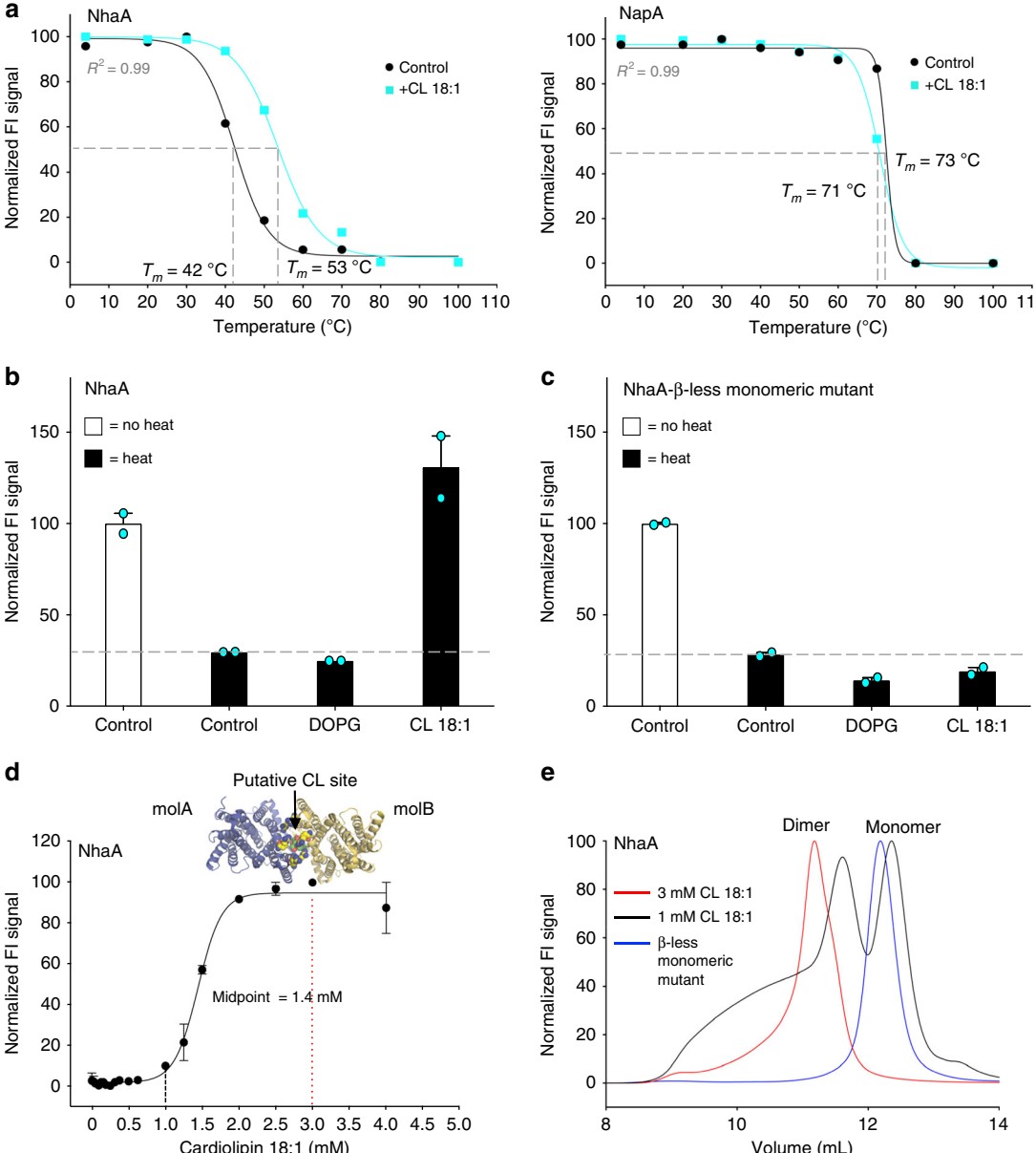

**Fig. 1** Validation of FSEC-TS for measuring specific lipid–protein interactions. **a** FSEC-TS melting curves for the purified sodium/proton antiporters NhaA-GFP (left) and NapA-GFP (right) in the absence (black) and presence of cardiolipin (CL) (cyan). Apparent $T_m$ of crude-detergent solubilized bacterial membrane protein fusions that were determined by the FSEC-TS assay using a range of nine different temperatures that were fitted to a sigmoidal dose–response equation as described in Methods. Each apparent $T_m$ (mean ± s.e.m. of the fit) is the average from two independent experiments. **b** FSEC peak fluorescence of purified NhaA-GFP before heating (open bars) and that remaining after heating and centrifugation (black bars) in presence of either CL or DOPG; error bars show the range of two technical replicates. **c** As in **b** for the monomeric NhaA-β-less mutant. **d** CL dependent stabilization of NhaA as assessed by FSEC-TS. Black and red lines indicate the preparations shown in **e**; inset shows the putative CL binding site between NhaA positively charged promoter surfaces in the dimeric crystal structure (pdb ID 4ATV); $n = 3$ independent titrations and the error bars show the mean ± s.d of the fit. **e** Normalized FSEC-TS traces from the CL titration to NhaA shown in **d** with either 1 mM CL (black) or 3 mM CL (red), and the NhaA-β-less monomeric mutant (blue)

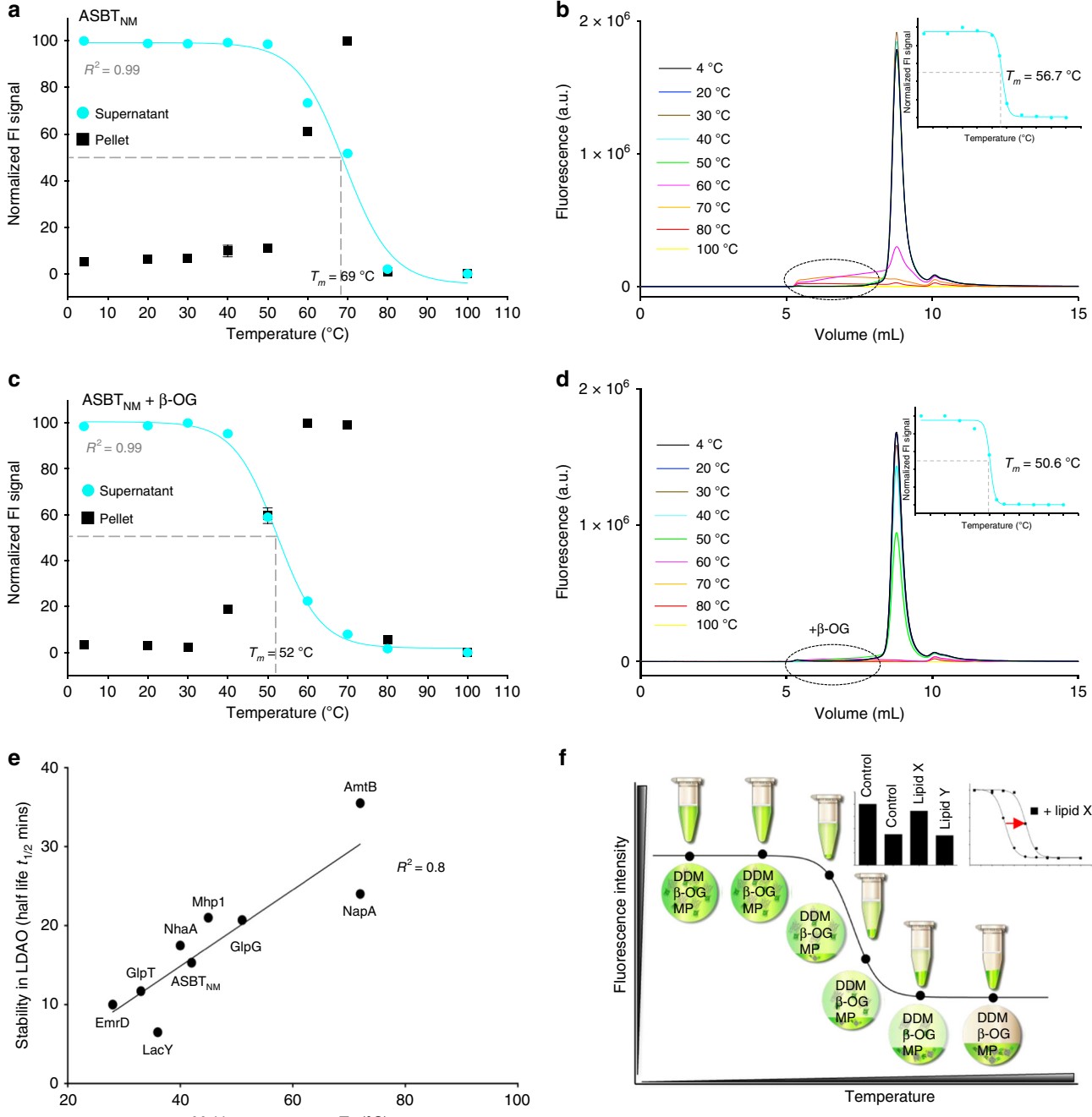

**Fig. 2** Developing GFP-TS for high-throughput lipid screening. **a** Total fluorescence, measured on a 96-well microplate spectrofluorometer, from DDM-solubilized ASBT$_{NM}$-GFP membranes remaining in solution after heating and centrifugation (cyan circles). Fluorescence from pellets resuspended in the same volume of buffer as the supernatant (black squares). Apparent $T_m$ were calculated using a range of nine different temperatures that were fitted to a sigmoidal dose–response equation as described in Methods (bars show the range of two technical replicates, and the values reported are the mean ± s.e.m. of the fit). **b**. FSEC-TS traces of the ASBT$_{NM}$-GFP supernatants measured in **a** with a plate reader; the aggregates are highlighted by a dotted circle and the corresponding FSEC-TS melting curve as an inset **c**. As in **a** except after the addition of the detergent β-OG to the DDM-solubilized ASBT$_{NM}$-GFP samples; error bars show the range of two technical replicates and the values reported for the apparent $T_m$ are the mean ± s.e.m. of the fit, which were further combined with a separate titration and are tabulated in Table 1. **d** FSEC-TS traces of the ASBT$_{NM}$-GFP supernatants measured in **c** with a plate reader; the location of the aggregates removed by β-OG are highlighted by a dotted circle and the corresponding FSEC-TS melting curve as an inset. **e** Correlation between the thermostability assessed by the direct CPM assay on DDM-purified fusions in the detergent LDAO[21] versus the indirect GFP-TS assay on unpurified fusions in the detergents DDM and β-OG; the transporters listed are also described in Table 1. **f** Schematic representation of the GFP-TS assay

**Table 1 Membrane bilayers are made up of a myriad of different lipids that affect membrane proteins, but identifying those specific lipid requirements remains a challenge. Here, authors present an engineered thermal-shift screen which reveals specific lipid preferences of eukaryotic and prokaryotic membrane proteins**

| Name | Function | Melting temperature, $T_m$ (°C) FSEC-TS ($+\beta$-OG) | Melting temperature, $T_m$ (°C) GFP-TS ($+\beta$-OG) | $T_m$ difference (°C) |
|---|---|---|---|---|
| AmtB | Ammonium channel | 77.4 ± 0.4 | 77.9 ± 0.3 | +0.5 |
| ASBT$_{NM}$ | Bile acid transporter | 50.6 ± 0.7 | 52.6 ± 0.4 | +2.0 |
| EmrD | Multidrug transporter | 32.7 ± 0.8 | 34.8 ± 0.7 | +2.1 |
| GlpG | Rhomboid protease | 61.7 ± 1.5 | 63.9 ± 0.5 | +2.2 |
| GlpT | Glycol-3-phosphate transporter | 33.3 ± 0.4 | 35.9 ± 0.7 | +2.6 |
| Mhp1 | Benzyl-hydantoin transporter | 37.5 ± 0.5 | 41.0 ± 0.3 | +3.5 |
| LacY | Lactose permease | 33.1 ± 0.1 | 35.7 ± 0.3 | +2.6 |
| NapA | Sodium/ proton antiporter | 70.6 ± 0.2 | 72.1 ± 0.9 | +1.5 |
| NhaA | Sodium/proton antiporter | 38.4 ± 1.1 | 42.6 ± 1.2 | +3.9 |

transporter), LacY (lactose symporter), NapA (sodium/proton antiporter), and NhaA (sodium/proton antiporter) (Table 1, see Methods). Again, each apparent $T_m$ was calculated from the recordings of the fluorescent material, prior to analysis by HPLC, and subsequently from the peak heights of the resulting FSEC-TS traces (Supplementary Figs. 2, 3). As shown in Table 1 and Supplementary Fig. 4, the apparent $T_m$ values calculated with or without the size-exclusion steps are in good agreement. In contrast, the apparent $T_m$ values in the absence of β-OG addition diverges considerably for all membrane proteins except for the exceptionally stable membrane proteins AmtB and NapA (apparent $T_m > 70$ °C), which do not aggregate under these conditions (Supplementary Fig. 1b). The addition of β-OG to the DDM-solubilized proteins only decreased the overall apparent $T_m$ by ~5 °C, which suggests that these experimental parameters should be suitable for most membrane proteins (Table 1 and Supplementary Fig. 1b).

Using the N-[4-(7-diethylamino-4-methyl-3-coumarinyl) phenyl] maleimide (CPM) thermal-shift assay, we previously reported the thermostabilities of purified, detergent-solubilized membrane proteins without the GFP-tag[21], which we now measured in detergent-solubilized membrane fractions using our modified assay. In principle, the thermostability assessments from the detergent-solubilized membrane fractions using our modified assay should be comparable to the CPM method using purified protein, and we indeed find a strong correlation made by these direct vs. indirect methods, $R^2 = 0.8$ (Fig. 2e). Notably, the modified assay also showed a similar midpoint for the titration of cardiolipin (18:1) stabilized dimerization of NhaA in detergent as that assessed by FSEC-TS (Supplementary Fig. 1c). Taken together, we conclude that the apparent $T_m$ of a membrane protein in detergent-solubilized membranes can easily be determined from the fluorescence signal of detergent-solubilized GFP-fusions remaining in the solution after heating and centrifugation. Since a size-exclusion chromatography step is no longer required, we refer to this methodology as the GFP-TS (Fig. 2f).

**Thermostabilities of pro- and eukaryotic membrane proteins**. Empirically, eukaryotic membrane proteins are reported to be less stable in detergent solution than their bacterial counterparts[21]. Thus far, however, and to the best of our knowledge, technical bottlenecks have hindered the systematic quantification of thermostability differences between bacterial and eukaryotic membrane proteins in a membrane environment. As membrane protein stability is a critical parameter towards their functional and structural characterization[22,23], we applied our GFP-TS assay to this complex biological problem.

Membranes were individually isolated from *S. cerevisiae* cells producing seven different eukaryotic solute carrier (SLC) transporters of therapeutic interest (Supplementary Table 1). Membrane solubilization was also carried out with 1% (w/v) DDM at a final total protein concentration of 3.5 mg mL$^{-1}$, except for two transporters where ~7.0 mg mL$^{-1}$ had to be used because of low-expression levels, i.e., to ensure the fluorescence would be high enough for accurate measurements (see Methods). To ensure this difference would not influence the stability measurements we confirmed that the apparent $T_m$ as determined by the GFP-TS assay was almost equivalent over this solubilization range (Supplementary Fig. 1d). Comparing the stabilities of prokaryotic and eukaryotic membrane proteins in membrane isolates revealed somewhat unexpectedly that the median apparent $T_m$ of the bacterial membrane proteins at 42.3 °C was only ~5 °C higher than the eukaryotic SLC transporters at 37 °C (Fig. 3a and Supplementary Figs. 5–7). Subsequently we repeated the GFP-TS assay on the detergent-solubilized and purified membrane proteins. The apparent $T_m$ for the bacterial membrane protein stability remained similar at 41.7 °C. However, the median apparent $T_m$ of the eukaryotic transporters was now 25 °C, which was substantially lower than the stability measured in the crude detergent-solubilized membranes (Fig. 3a and Supplementary Figs. 5–7). To confirm that the ~$\Delta T_m$ 17 °C stability differences observed between the purified eukaryotic and bacterial membrane proteins was not simply a consequence of different overall structures, we focused on the monosaccharide proton-coupled transporter XylE from *E. coli*, as the available crystal structures display a high degree of structural similarity to the mammalian glucose GLUT transporters GLUT1 and GLUT5[24,25] (Fig. 3b). In detergent-solubilized membranes, the apparent $T_m$ of bacterial XylE was determined as 42 °C, which was similar to human GLUT1 at 39 °C, and ~9 °C higher than rat GLUT5 at 33 °C. After purification, the apparent $T_m$ of XylE remained at 41°C, while hGLUT1 and rGLUT5 dropped substantially to 27 and 23 °C, respectively, mirroring closely the trend of our overall test-set (Fig. 3b, c and Supplementary Figs. 5–7).

The similar apparent $T_m$ of the unpurified membrane proteins from the yeast and bacterial membranes indicates protein stability is not overly biased by the respective expression hosts (Supplementary Table 1). Nonetheless, to exclude the possibility that *E. coli* lipids are not generally better at stabilizing bacterial membrane proteins than *S. cerevisiae* lipids or vice versa, we supplemented crude membranes containing GLUT5 or XylE with either detergent-solubilized *E. coli* or *S. cerevisiae* membranes, respectively, and measured their stabilities before and after purification. Even when starting with the same total pool of lipids, however, similar changes in the apparent melting temperatures

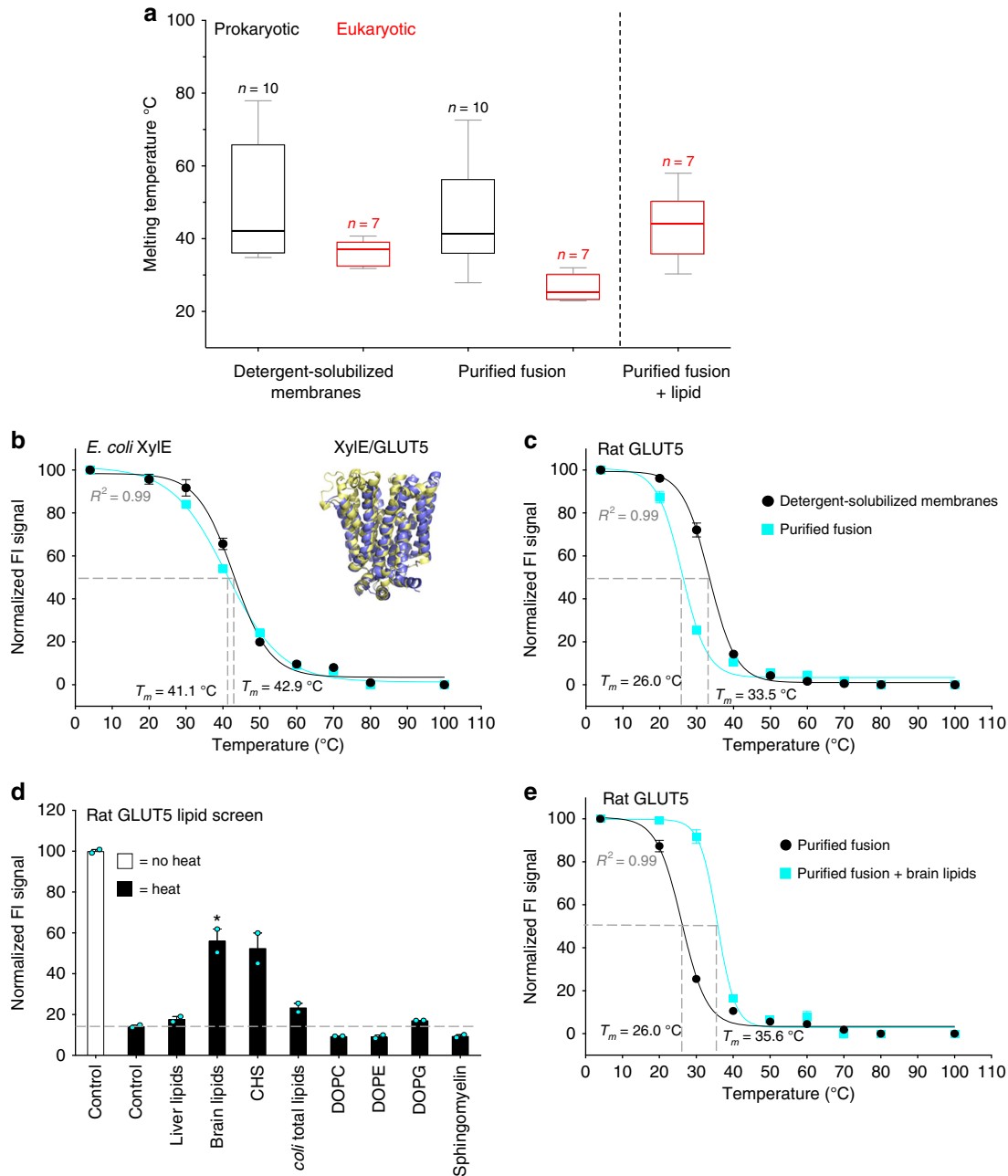

**Fig. 3** Comparing the lipid stabilization of bacterial and eukaryotic membrane proteins. **a** Box-and-whisker plots show the distribution of thermostabilities for eukaryotic membrane proteins (red bars) and bacterial membrane proteins (black bars) before and after purification as assessed by the GFP-TS assay; the median is shown as a line in the box, while bottom and top boundaries represent the lower and upper quartile, respectively. Whiskers indicate the minimum and maximum apparent $T_m$. The individual apparent $T_m$ values are listed in Supplementary Table 1. **b** The GFP-TS melting curves for the bacterial monosaccharide transporter XylE in crude-detergent solubilized membranes (black circles) and as a purified fusion (cyan squares); inset shows the structural similarity between GLUT5 (pdb ID 4YBQ) and XylE (pdb ID 4GBY) crystal structures; error bars show the range of two technical replicates and the values reported for the apparent $T_m$ are the mean ± s.e.m. of the fit, which were further combined with a separate titration and are tabulated in Supplementary Table 1. **c** As in **b** for the mammalian monosaccharide transporter rat GLUT5. **d** Supernatant fluorescence of purified rat GLUT5-GFP before heating at apparent $T_m + 5\,°C$ (nonfilled bars) and that remaining after heating and centrifugation (black bars) in presence of listed lipids; the asterisk indicates the most stabilizing lipid (bars show the range of two technical replicates) **e** GFP-TS melting curves for purified rat GLUT5 in the absence (black) and presence of brain lipids (cyan); apparent $T_m$ were calculated as described in **b** and the values reported are the mean ± s.e.m. of the fit (see also Supplementary Table 1)

were observed, with only GLUT5 destabilized after purification (Supplementary Fig. 8). It is still possible that the *E. coli* lipids are a closer match to the native lipid composition of the bacterial proteins and are, therefore, better at stabilizing the bacterial proteins than the *S. cerevisiae* lipids are at stabilizing the

mammalian proteins. To test for this, we isolated membranes from human embryonic kidney (HEK) cells and added them in equimolar amounts to detergent-solubilized *S. cerevisiae* membranes expressing human NHA2, which is natively expressed in the kidney[26]. We repeated the stability analysis of human NHA2,

but again found that the apparent $T_m$ also dropped substantially after purification (Supplementary Fig. 8). Taken together, our data suggest that eukaryotic membrane proteins may not necessarily be more unstable than bacterial membrane proteins in their native membranes, but they appear to be more sensitive to detergent solubilization and lipid removal than their bacterial counterparts.

**Lipid stabilization of purified eukaryotic membrane proteins.** During the purification of a membrane protein in detergent solution, lipids are progressively lost[27]. It was well-known that some eukaryotic membrane proteins require the addition of lipids during purification to avoid aggregation[28–30]. If the loss of lipids is the foremost reason for their destabilization in detergent, it should, therefore, be possible to reverse the negative effect by adding specific lipids back. To screen for lipid stabilization, we selected a range of different lipids that are highly abundant in bacterial and mammalian membranes and added them to the purified GFP fusion proteins prior to heating at a temperature 5°C above their individual apparent $T_m$ (Fig. 3a, d and Supplementary Figs. 9, 10).

Among the eukaryotic membrane proteins, the addition of *E. coli* total lipids, which consists of PE, PG, and cardiolipin, had little or no effect on their thermostability (Fig. 3d and Supplementary Figs. 9, 10). The addition of individual PG or PE lipids yielded similar results, however, not in all cases, such as human NHA2, where the addition of PG was clearly stabilizing. Overall, however, bacterial membrane lipids do not seem to substantially increase protein stability, which is consistent with the inability of *E. coli* lipids to stabilize rat GLUT5 during purification (Supplementary Fig. 8). In comparison, the addition of cholesteryl hemisuccinate (CHS) or lipids from bovine brain provided pronounced stabilization effects in most cases. Calculating the apparent $T_m$ of all the eukaryotic membrane proteins in the presence of their most stabilizing lipid revealed higher average stabilities than in crude yeast membrane fractions, and similar to that of the bacterial membrane proteins in bacterial membrane fractions (Fig. 3a, e and Supplementary Table 1).

**GFP-TS to aid functional–structural studies of transporters.** Many membrane proteins, such as those belonging to the SLC family of transporters, have no known function[31]. In vitro functional assays are the gold-standard, but can be technically difficult to establish and are often hindered by the unavailable of suitable radiolabeled ligands for biochemical assays[31,32]. We, therefore, decided to evaluate if the GFP-TS could also encompass the screening of small molecule ligands to aid the functional characterization of membrane proteins. As a test-case, we screened possible substrates to detergent solubilized membranes harboring an uncharacterized plant transporter from *Zea mays* that shares ~28% sequence identity to the human nucleotide-sugar transporter (SLC35A1), which transports CMP–sialic acid into the ER and Golgi lumen in exchange for CMP for sialylation of secretory proteins[33,34]. In this case, it was unclear if CMP–sialic acid was really a substrate for the plant protein as nucleotide-sugar transporters can have overlapping substrate specificities and even members with high sequence identity can transport different substrates[32,34]. Using GFP-TS we could however observe clear and specific thermostabilization upon the addition of CMP, which is a counter-substrate unique to the nucleotide-sugar transporters that transport CMP–sialic acid (Fig. 4a). Thermostabilization was ~eightfold higher in contrast to closely related UMP or GMP nucleoside monophosphates, which are counter-substrates for nucleotide-sugar transporters translocating UDP- and GDP-linked sugars, e.g., UDP–glucose.

We further observed higher thermostabilization for CMP–sialic acid as compared to the addition of sialic acid only. Moreover, an almost identical stabilization profile for substrates was observed for the human CMP–sialic acid transporter SLC35A1, further supporting the identification of the correct substrate (Fig. 4a). From unpurified detergent solubilized membranes, we used the GFP-TS to determine CMP binding affinities ($K_d$) for the plant and human transporters, which were found to be consistent to each other and also to isothermal titration colorimetry (ITC) measurements from purified proteins (Fig. 4b and Supplementary Fig. 11a). Thus, the GFP-TS assay can be used for identifying potential substrates of eukaryotic transporters using unlabeled compounds and small amounts of unpurified proteins.

We next asked whether the stabilizing effects of ligands observed in the GFP-TS assay could be harnessed to also aid the crystallization of membrane proteins. Membrane protein crystals grown by in meso or the lipidic cubic phase (LCP) method generally produce higher resolution structures, as they have a lower solvent content (type I crystals) than those grown by traditional vapor-diffusion crystallization (type II crystals). To grow LCP crystals of membrane proteins with the synthetic lipid monoolein, the purified membrane protein solution is mixed with the molten monoolein in a weight ratio of 2:3[35]. It can be very challenging to grow LCP crystals of membrane proteins, however, and while it is generally thought to be a fairly mild environment, the stabilities of different membrane proteins have not been extensively compared.

Using the GFP-TS assay we added monoolein to the purified membrane proteins at a concentration range used for LCP crystallization. Interestingly, the monoolein lipid was in fact very destabilizing for most of the eukaryotic membrane proteins tested (Supplementary Fig. 11b). Indeed, despite many attempts we were not able to grow LCP crystals for any of these eukaryotic SLC transporters. To see if the destabilizing effects of monoolein could be compensated for by the addition of a stabilizing ligand, we repeated the analysis of NhaA with and without the addition of the stabilizing lipid cardiolipin. Indeed, the apparent $T_m$ of NhaA was significantly decreased in the presence of monoolein (from 42 to at 27 °C), but was rescued in the presence of cardiolipin (41 °C) (Fig. 4c, d). We repeated the analysis for the plant CMP–sialic acid transporter and likewise found that the apparent $T_m$ was significantly decreased in the presence of monoolein (from 48 to at 40 °C), but rescued in the presence of CMP (50 °C) (Fig. 4e, f). As proof-of-principle of this approach, we could now obtain well-diffracting LCP crystals of both NhaA and the plant CMP–sialic acid transporters when supplemented with these ligands (Supplementary Fig. 11c, d). Taken together, we show that the GFP-TS assay can be used as a simple screening tool to identify key ligands for functional and structural characterization of membrane proteins.

## Discussion

The bilayer is an intricate network of many lipids with different biophysical and biochemical properties, but the mechanisms by which specific lipids can fine-tune the activity of membrane proteins are largely unknown. Recent data suggest that point mutations in membrane proteins, such as GPCRs, may even affect lipid interactions that result in an altered drug response[36]. Despite their importance, simple tools for quantifying lipid interactions have been lacking and this has hindered the routine evaluation of how lipids might be influencing membrane protein stabilization, oligomerization and function, and how these factors may affect disease mechanisms.

In the present study, we have demonstrated that FSEC-TS is suitable to screen and measure specific lipid interactions with

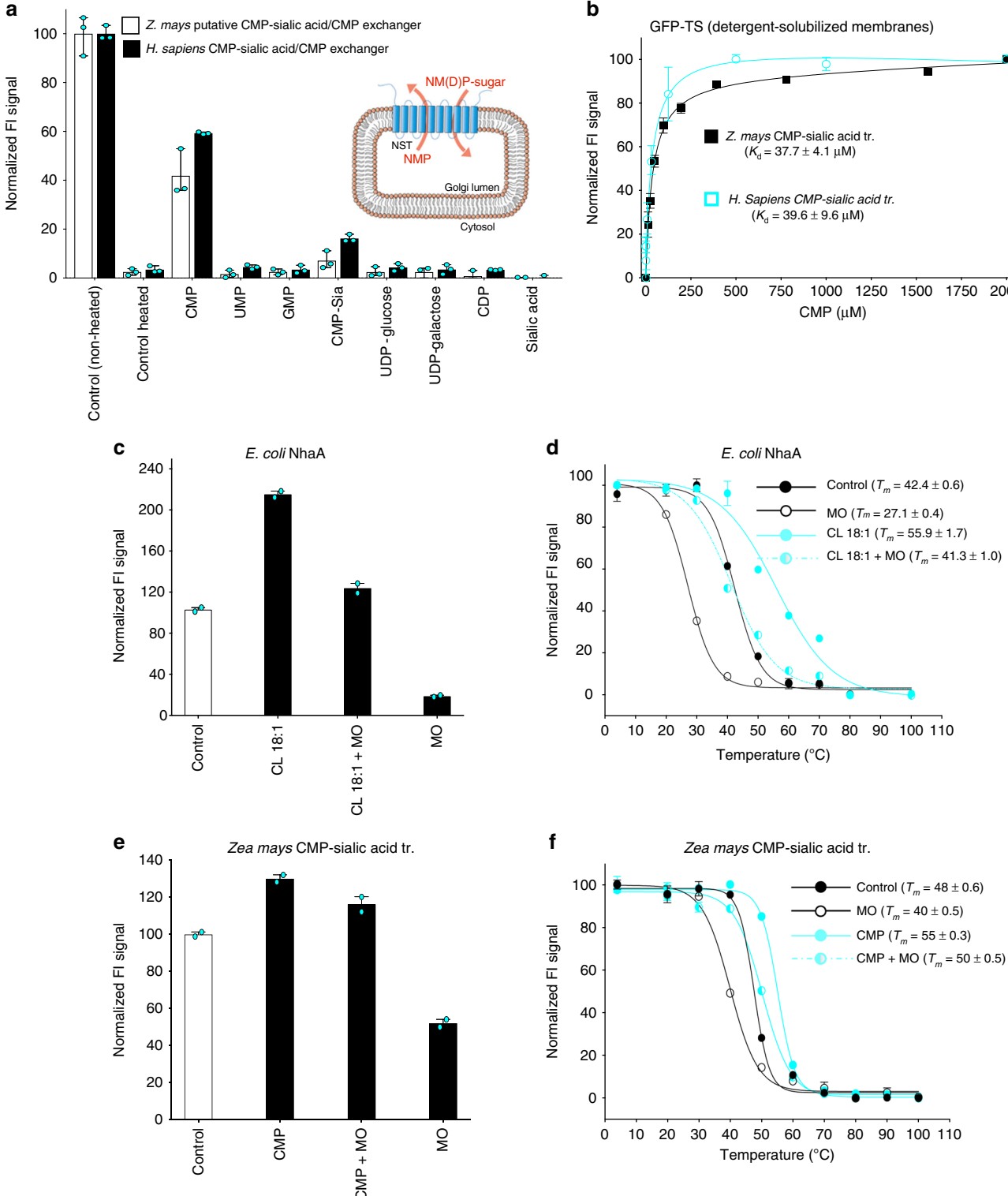

both bacterial and eukaryotic membrane proteins. Using FSEC-TS we are able to capture the specific role of cardiolipin (18:1) in stabilizing the dimeric architecture of the sodium/proton antiporter NhaA as was previously indicated by MS and MD simulations[14,15]. Sodium/proton antiporters are known to form dimers under physiological conditions[37], and it has been shown that the dimeric form of NhaA is required for *E. coli* to grow under conditions of high-salt stress[18,38]. Considering that under high-salt conditions, where NhaA activity is upregulated[38], the

cardiolipin content of *E. coli* membranes also increases[39], the cardiolipin-mediated tuning of NhaA dimerization shown by FSEC-TS implies a potential regulatory mechanism with wide-ranging physiological ramifications.

To allow application of the powerful FSEC-TS methodology[13] to a wider range of targets and environments, we have removed the requirement of a size-exclusion chromatography step in the GFP-TS assay. Importantly, the thermostability measurements made by the resulting GFP-TS in crude-membranes correlated

**Fig. 4** Using GFP-TS for functional and structural characterization of membrane proteins. **a** Supernatant fluorescence of detergent solubilized membranes containing either a putative *Zea mays* CMP–sialic acid transporter (nonfilled bars) or the human CMP–sialic acid transporter (filled bars) before heating and that remaining after heating ($T_m + 4 °C$; human and $T_m + 6 °C$; *Z. mays*) and centrifugation in presence of listed ligands (bars show the range of three technical replicates). inset; schematic of a Golgi nucleotide-sugar transporter (NST) that exchangers an NM(D)P-sugar for NMP. **b** Binding affinity ($K_d$) of CMP to the putative *Z. mays* CMP–sialic acid transporter (black bars) and the human CMP–sialic acid transporter (cyan bars) as assessed by GFP-TS using detergent solubilized membranes; $n = 3$ independent titrations and the error bars show the mean ± s.d of the fit. **c** Effect of monoolein on the stability of NhaA. The stability of NhaA was restored by addition of cardiolipin (18:1) in the presence of monoolein which destabilizes NhaA (error bars show the range of two technical replicates). **d** Apparent $T_m$ measurement of NhaA with no addition of lipid (black, closed circle), addition of monoolein (black, open circle), addition of cardiolipin 18:1 (cyan, closed circle) and addition of both monoolein and cardiolipin 18:1 (cyan, semi-closed circle). **e** Effect of monoolein on the stability of *Z. mays* CMP–Sia. The stability of *Z. mays* CMP–Sia was restored by addition of CMP in the presence of monoolein which destabilizes *Z. mays* CMP–Sia (error bars show the range of two technical replicates). **f** GFP-TS melting curves for purified *Z. mays* CMP–sialic acid transporter in the absence (black; filled circles) and presence of either monoolein lipid (black; nonfilled circles) or CMP (cyan; filled circles) or both monoolein and CMP (cyan; half-filled); error bars show the range of two technical replicates and the values reported for the apparent $T_m$ are the mean ± s.e.m. of the fit

well with the stability estimates in detergent solution from purified samples using the CPM assay, which monitors unfolding of the protein directly[11]. Although it is possible that the parameters presented here may need to be further fine-tuned to be consistent with the stability determined by FSEC-TS in some cases, once optimized, GFP-TS is also amenable in a 96-well format, which should also facilitate its use for the high-throughput screening of small molecule libraries. An obvious advantage compared to thermal-shift assays using thermofluor dyes is that the membrane proteins do not need to be purified first, and potential interactions can be screened in the presence of their native lipids, which can be critical for the reliable identification of in vivo ligands[40]. Indeed, we could use the GFP-TS assay to establish the substrate specificity of a plant nucleotide-sugar transporter homolog of human SLC35A1, which belongs to a family that despite its role in human diseases, to date, has proved a considerable challenge to characterize[32]. Indeed, many SLC transporters have no known function, which is an important issue since close to half of them are associated with human disease and are important drug targets[31,41].

Probably one of the most important parameters in the structure determination of membrane proteins, by either crystallography or single-particle cryo EM, is their stability in detergent solution[22,23]. One of the most widely used approaches for screening homologs and optimizing membrane proteins for structural investigation is to couple their production with a C-terminal GFP tag[42,43] The cleavable GFP-tag enables a fast readout of expression levels[42,43] integrity by in-gel fluorescence[43] and monodispersity of the protein–detergent complex by FSEC[44]. GFP-based membrane protein screening platforms have been well established in *E. coli*[45] yeast[46,47], insect[48], and mammalian cells[49]. Thus, the extension of the GFP-based methodology should make the screening for specific lipids or other membrane protein ligands to in a high-throughput format easily applicable from a variety of expression hosts, and further aid functional and structural determination, as demonstrated here for the Na$^+$/H$^+$ exchanger NhaA and the eukaryotic CMP–sialic acid transporter.

We find it particularly noteworthy that before purification, the eukaryotic membranes proteins in our test-set appear to be almost as stable as the bacterial membrane proteins and are only destabilized as a result of the delipidation process. By screening a handful of different lipids, we have shown a striking potential for the targeted stabilization of purified eukaryotic membrane proteins. While most stabilizing strategies of mammalian membrane proteins, such as GPCRs, have focused on the screening of stabilizing point mutations[23,50], we find it remarkable that there are subtle differences yet to be established at the structural level that govern the sensitivity toward the presence of certain lipids. It is likely that the destabilization of membrane proteins after moving from lipids to detergent micelles is a consequence of increased

conformational dynamics, which is consistent with the observations made for GPCRs[51]. Some lipids, therefore, might be required to conformationally stabilize membrane proteins, and mammalian membrane proteins may have evolved to be more sensitive toward specific environments that promote or reduce conformational flexibility. The stabilization seen for cholesteryl hemisuccinate, which is thought to be interact with membrane proteins in a similar manner to cholesterol[52] is consistent with this rationale, since growing evidence indicates that cholesterol preferentially interacts with membrane proteins to restrict dynamics[53,54]. Given that dynamics and function are intimately entwined, this further implies that the greater sensitivity of the eukaryotic proteins to their environmental lipid composition might be connected with their clear ability to be allosterically regulated by these lipids (e.g., [55–57]).

In summary, although we are just beginning to dissect how specific lipids interact with membrane proteins to modulate their activity, we think that the methodology and the resulting insights present an approach for investigating and establishing the important roles of lipids and other ligands with their interplay with membrane proteins. In particular, the strategy presented complements with MS to identify stabilizing lipids, ligand, and drug interactions in native environments for in-depth analysis of the functional and structural implications.

## Methods

**Isolation of membrane protein–GFP fusion membranes**. Bacterial membrane protein–GFP fusions: The bacterial membrane protein–GFP$_{8His}$ fusions had been previously constructed into the expression vector, pWaldo GFPd[21,43]. For overexpression, a single colony of freshly transformed *E. coli* Lemo21 (DE3) cells from New England Biolabs harboring a pWaldo membrane protein–GFP$_{8His}$ fusion on a Luria Bertani (LB) agar/kanamycin plate, was used to inoculate 20 mL of LB broth supplemented with 50 μg mL$^{-1}$ kanamycin and grown at 37 °C with shaking at 200 rpm using Innova 44 Incubator shaker (NEW BRUNSWICK) for 16 h. The cultures were then used to inoculate 2 × 1 L of MemStar medium[45] and grown to an OD$_{600}$ of 0.5. The cells were then induced with 0.4 mM IPTG (Sigma) and the temperature of the shaker dropped from 37 to 25 °C and grown further for 16 h. Cells were harvested by centrifuging at 10,000*g* for 10 min at 4 °C, resuspended in 200 mL of ice-cold 1× phosphate-buffered saline (PBS) buffer and flash-frozen in liquid nitrogen before storing at −80 °C until use. Membranes containing membrane protein–GFP$_{8His}$ fusions were isolated by breaking the cells using a constant cell disruption system (two 25 kpsi passes at 4 °C). Unbroken cells and debris were removed by centrifugation at 10,000*g* for 10 min at 4 °C. The supernatant containing membranes was then centrifuged at 150,000*g* for 1 h and the pellets resuspended in 20 mL of ice-cold 1× PBS buffer. Total membrane protein concentration was measured using the BCA assay (Pierce BCA Protein Assay Kit, Thermo Fischer Scientific), and the resuspended membranes were flash-frozen in liquid nitrogen and stored at −80 °C until use.

Eukaryotic membrane protein–GFP fusions: The cDNA of mouse NHA2 (UniProt: Q5BKR2), bird FLVCR (A0A0Q3MM10) and *Oryza sativa* CMP–Sia (Q654D9) were synthesized and amplified by PCR using forward 5′-ACCCCGGATTCTAGAACTAGTGGATCCCCC-3′ and reverse 5′-AAATTGACCTTGAAAATATAAATTTTCCCC-3′ primers and cloned by homologous recombination into the SmaI linearized vector pDDGFP-2 during transformation into the *S. cerevisiae* strain FGY217 (MATα, *ura3*–52, *lys2*Δ201,

*pep4Δ*[46,47]. The clone for *Z. mays* CMP–Sia (B4FZ94) was synthesized and for human GLUT1 (P11166)[58], rat GLUT5 (P43427)[25], human NHA2 (Q86UD5)[59], and human CMP–Sia (P78382)[47] were generated previously into the pDDGFP-2 vector and were likewise transformed into the FGY217 strain[47]. Overexpression and membrane isolation of the eukaryotic membrane proteins from 12L cultures were carried out according to our published protocol[46].

**Purification of membrane protein–GFP fusions.** Isolated membranes from either 2L of *E. coli* cultures or 12L of *S. cerevisiae* cultures harboring the separately overexpressed GFP-fusions were diluted to a total protein concentration to 3.5 mg mL$^{-1}$ (final) in buffer containing 1× PBS, 150 mM NaCl and 1 % (w/v) DDM. After 1 h gentle mixing at 4 °C for 1 h the material was spun at 120,000$g$ at 4 °C for 45 min to remove the unsolubilized material. A 10 mM imidazole pH 7.5 was added to the cleared supernatant and incubated with either 5 mL (bacterial proteins) or 10 mL (eukaryotic proteins) of Ni$^{2+}$-nitrilotriacetate affinity resin (Ni-NTA; Qiagen) for 2 h at 4 °C. The resin was washed with 20 column volumes (CVs) of buffer containing 1× PBS, 150 mM NaCl, 0.1% (w/v) DDM and 30 mM imidazole pH 7.5. The protein eluted in 3 CVs of 1× PBS, 150 mM NaCl and 0.1% (w/v) DDM, and 250 mM imidazole pH 7.5 and dialyzed o/n in 3L buffer containing 20 mM Tris-HCl pH 7.5, 150 mM NaCl, and 0.03% (w/v) DDM. To determine yields, 100 µL of the purified fusion was transferred to a 96-well black optical bottom plate and the GFP fluorescence (RFU) measured using a microplate spectrofluorometer with excitation 488 nm and emission 512 nm. GFP fluorescence was converted into amounts of purified fusion[46]. The qualities of all prokaryotic and eukaryotic membrane proteins were assessed to be monodisperse by FSEC (as described in next section). All purifications were repeated at least twice. With the exception of XylE, all of the prokaryotic membrane purified using our standard procedure could be crystallized for structural investigation[17,20,21,60]. Detailed structure and/or functional activities were also confirmed for purified rat GLUT5[25], human GLUT1[58], and human NHA2[59], and we were further able to establish functional transport activities for *E. coli* XylE, mouse NHA2, plant CMP–Sia, and human CMP–sialic acid transporters.

**GFP-TS and FSEC-TS measurements.** Membranes were diluted into 5 mL of buffer containing 150 mM NaCl, 20 mM Tris-HCl pH 7.5 and 1% (w/v) DDM to a total membrane protein concentration of ~3.5 mg mL$^{-1}$, except for hGLUT1 and FLVCR2 that had a final total protein concentration of ~7 mg mL$^{-1}$ to ensure the fluorescence signal would be high enough for accurate measurements (>5000 RFU). Membranes were solubilized for 1 h with mild agitation at 4 °C. For GFP-TS measurements, and where otherwise stated, 1% (w/v) β-OG was further added to the solubilized membranes or 3.5 mL of purified fusions that had been adjusted to 5000 RFU in buffer containing 150 mM NaCl, 20 mM Tris-HCl pH 7.5 and 0.03% (w/v) DDM that corresponds to a final protein concentration of ~0.02–0.04 µg µL$^{-1}$[46].

Totally, 150 µL of either solubilized membranes or purified fusions were subsequently transferred into 1.5 mL tubes for 10 min at 4, 20, 30, 40, 50, 60, 70, 80, and 100 °C using a Thermomixer Comfort heating block (Eppendorf) without mixing. The heated samples were then pelleted at 18,000$g$ for 30 mins at 4 °C using a Microfuge® 18 Centrifuge, Beckman Coulter. For GFP-TS, the supernatants were transferred into a 96-well black clear bottom plate (Nunc) and the GFP fluorescence (excitation set at 488 nm and emission at 512 nm) measured using a SpectraMax Germini EM microplate reader (Molecular Devices), which takes ~1 min. The apparent $T_m$ for each titration was calculated by plotting the average GFP fluorescence intensity from two technical repeats at each temperature and fitting the curves to a sigmoidal dose–response equation by GraphPad Prism software. Two technical repeats were considered sufficient for accurate $T_m$ calculations as the goodness of the fit was >0.98 and the range between two technical repeats very low. The tabulated apparent $T_m$ (mean ± s.e.m. of the fit) shows the average from two independent detergent-solubilization and membrane protein purifications.

After recording the GFP fluorescence from the microplate spectrofluorometer, for FSEC-TS, 100 µL of each sample was recovered and further injected using an autosampler onto either a BioSep™ 5 µ5 SEC-s3000 400 Å, LC Column (Phenomenex, Part No. 00H-2146-K0) or ENrich™ SEC 650 10 × 300 Column (BIO-RAD pre-equilibrated with 20 mM Tris-HCl pH 7.5, 150 mM NaCl, 0.03% (w/v) DDM at a flow rate of 1 mL min$^{-1}$ at room temperature using an inline-detector Shimadzu HPLC system (Shimadzu Corporation)). The total time for each FSEC-TS titration was ~3–4 h. The apparent FSEC-TS $T_m$ was subsequently obtained by plotting the GFP fluorescence peak intensity from single measurements at each temperature and fitting the curves to a sigmoidal dose–response equation by GraphPad Prism software[13]. The tabulated apparent $T_m$ (mean ± s.e.m. of the fit) was obtained after averaging two independent detergent-solubilizations and purifications.

**Purification of fusions with crude membrane mixtures.** *E. coli* and *S. cerevisiae* membranes were isolated from empty cells in the same manner as those harboring GFP-fusions. Empty *E. coli* and *S. cerevisiae* membranes were diluted to 3.5 mg mL$^{-1}$ in buffer containing 150 mM NaCl, 20 mM Tris-HCl pH 7.5 and added to rat GLUT5 *S. cerevisiae* containing membranes or *E. coli* XylE containing *E. coli* membranes at 3.5 mg mL$^{-1}$ in the same buffer, respectively. Mixed membranes

containing *E. coli* XylE or rat GLUT5 were solubilized in 1% (w/v) DDM for GFP-TS measurements as described in the previous section.

To assess if human lipids would be better at retaining the stability of human NHA2, membranes were first isolated from empty HEK cells. In brief, FreeStyle 293-F cells (Thermo Fisher Scientific) were cultured in a 2.8L Optimum Growth Flask (Thomson) at an initial density of $0.3 × 10^6$ cells/mL in FreeStyle 293 Expression Medium (ThermoFisher Scientific) at 37 °C in a humidified atmosphere with 8% $CO_2$, using an orbital shaker platform at 150 rpm. After 72 h, the cells were harvested by centrifugation at 1000$g$ for 5 min and subsequently resuspended in media containing 50 mM Tris-HCl (pH 7.5), 1 mM EDTA and 0.6 M sorbitol. Cells were lysed by sonication (Vibra-Cell VCX 130, Sonics) after three 10 s pulses at 80% amplitude. Unbroken cells and debris were removed by centrifugation at 10,000$g$ for 10 min at 4 °C and the membrane fraction was obtained by centrifugation at 150,000$g$ for 1 h. The membranes were resuspended in ice-cold 1× PBS and the total protein concentration was measured using the BCA assay. Empty human membranes were diluted to 3.5 mg mL$^{-1}$ in buffer containing 150 mM NaCl, 20 mM Tris-HCl pH 7.5 and added to human NHA2 *S. cerevisiae* containing membranes at 3.5 mg mL$^{-1}$ in the same buffer, respectively. Mixed membranes containing human NHA2 were solubilized in 1% (w/v) DDM for GFP-TS measurements as described in the previous section.

**Lipid screening of purified membrane protein–GFP fusions.** For lipid screening, the purified MP-GFP$_{8His}$ fusions were diluted in buffer containing 150 mM NaCl, 20 mM Tris pH 7.5 and 0.03% (w/v) DDM to a final GFP fluorescence count of at least 3000, which corresponds to ~0.01–0.03 mg mL$^{-1}$. A 96 µL was aliquoted into duplicate 1.5 mL tubes followed by the addition of 12 µL each of the following lipids (liver lipids (Avanti, cat. no. 181104P), brain lipid (Sigma cat. no. B1502), CHS (Sigma, cat. no. C6013), *E. coli* total lipids (Avanti, cat. no. 100500P), DOPC (Avanti cat. no. 850375P), DOPE (Avanti cat. no. 850725P), DOPG (Avanti cat. no. 840475P), sphingomyelin (Avanti cat. no. 860062P)) to a final concentration of 3 mg mL$^{-1}$. Stock solutions of lipids were prepared by solubilizing them in 10% (w/v) DDM to a final concentration of 30 mg mL$^{-1}$ overnight at 4 °C with mild agitation. Twelve microlitre of β-OG (10% (w/v) stock) was then added into the mixture to a final concentration of 1% (w/v) and then mixed by pipetting. For negative controls, 96 µL was aliquoted into four 1.5 mL tubes and 1% (w/v) DDM and 1% (w/v) β-OG (12 µL from 10% (w/v) stocks) were added. Purified fusions (apart from one of the negative controls) were heated for 10 min at 5 °C higher than their individual apparent $T_m$ and then centrifuged at 18,000$g$ at 4 °C using a Microfuge® 18 Centrifuge, (Beckman Coulter). The fluorescence from the supernatant was then measured on the microtitre plate spectrophotometer as described in the previous section.

**Cardiolipin 18:1-dependent oligomerization of *E. coli* NhaA.** Purified NhaA-GFP$_{8His}$ fusion protein was diluted in buffer containing 150 mM NaCl, 20 mM Tris pH 7.5 and 0.03% (w/v) DDM to a final RFU of ~5000, which corresponds to concentration of 0.02 µg µL$^{-1}$. A 120 µL of the diluted sample was then aliquoted into duplicate 1.5 mL tubes containing increasing concentrations of cardiolipin 18:1 (Avanti, cat. no. 710335P), as well as into a control containing an equal volume of 1% (w/v) DDM instead of lipid. β-OG was then added into all the mixtures to a final concentration of 1% (w/v) and then mixed by pipetting. Samples were heated for 10 min at 47 °C and centrifuged at 18,000$g$ using a Microfuge® 18 Centrifuge at 4 °C. The GFP fluorescence of the supernatant was then measured using a plate reader as described in the previous section, the samples recovered, and then further injected onto a ENrich™ SEC 650 10 × 300 Column (BIO-RAD) pre-equilibrated with 20 mM Tris-HCl pH 7.5, 150 mM NaCl, 0.03% (w/v) DDM at a flow rate of 1 mL min$^{-1}$ at room temperature using an inline-detector Shimadzu HPLC system (Shimadzu Corporation). The CL dependent oligomerization concentrations were calculated with 19 different cardiolipin (18:1) concentrations that were fitted by nonlinear regression, and the values reported are the averaged mean ± s.e.m. of the fit from $n = 3$ independent titrations.

**Substrate screening and binding affinity measurements.** For ligand screening, *S. cerevisiae* membranes containing either the human CMP–Sia transporter or the plant homolog from *Zea mays*, were diluted into buffer containing 150 mM NaCl, 20 mM Tris-HCl pH 7.5 and 1% (w/v) DDM to a final total membrane protein concentration of ~3.5 mg mL$^{-1}$ and solubilized for 1 h at 4 °C. The different ligands (CMP; Sigma C1006, CMP–sialic acid; SANYO FINE CO LTD 12595-31, UMP; Sigma U6375, CDP; Sigma C0256, Sialic acid; Sigma A0812, GMP; Sigma G8377, UDP–glucose; Sigma; 94335, UDP–galactose; Sigma U4500) were prepared in MilliQ water to a final concentration of 100 mM. A 6 µL of ligand was added to the membrane aliquots of the transporters to reach a final concentration of 5 mM and then mixed briefly by centrifugation. A 12 µL of 1% β-OG was subsequently applied to the respective supernatants at reach a final volume of 120 µL, mixed by brief centrifugation, and then heated for 10 min at either 54 °C (plant) or 42 °C (human), respectively using a Veriti 96-well thermal cycler (Thermo Fisher Scientific). These temperatures were empirically selected so that the fraction of nonheated to heated fluorescence was similar, that is, the level of heat-induced destabilization applied was equivalent. Fluorescent aggregates were pelleted at 18,000$g$ at 4 °C using a Microfuge® 18 Centrifuge, (Beckman Coulter) and the

supernatant fluorescence measured using a microtitre plate spectrophotometer, as described in the previous section.

For binding affinity measurements by GFP-TS solubilized membrane fusions were prepared as described for the ligand screening and ten different concentrations of CMP ranging from 0 to 2000 μM were added. The samples (apart from a negative control) were heated for 10 min at their individual apparent melting temperatures and subjected to centrifugation at 18,000g for 30 min at 4 °C using a Microfuge® 18 Centrifuge, (Beckman Coulter). The $K_d$ were calculated with ten different CMP concentrations that were fitted by nonlinear regression (one site, total binding), and the values reported are the averaged mean ± s.e.m. of the fit from $n = 3$ independent titrations

ITC measurements were made using the Micro-200 ITC (MicroCal, Malvern). Purified Z. mays CMP–Sia transporter and H. sapiens CMP–Sia transporters (from 12L) were concentrated to ~100 μL using 100 kDa MWCO concentrators. The final flow through containing buffer was used to dilute a 1 M stock solution of CMP to the final concentration used for measurements. The putative Z. mays CMP–sialic acid/CMP exchanger at 72 μM was loaded into the sample cell, and 1.2 mM CMP was loaded into the injection syringe while the H. sapiens CMP–sialic acid/CMP exchanger at 140 μM was loaded into the sample cell and 5 mM CMP was loaded into the injection syringe. The system was equilibrated to 20 °C with a stirring speed of 600 rpm. Titration curves were initiated by a 0.5 μL injection and were followed by 2 μL injections every 200 s. Background corrections were obtained by injecting CMP into buffer and buffer into protein with the same parameters. ORIGIN 7 was used to integrate, correct and normalize the heat for each injection and fit the data to a single-site binding isotherm with a fixed protein/ligand stoichiometry of 1, excluding the peak from the first injection.

**LCP crystallization of transport proteins**. NhaA purified with buffers containing of 2% (w/v) cardiolipin (18:1) (Avanti, cat. no. 710335P) was concentrated to 36 mg mL$^{-1}$ and used for in meso or lipidic cubic phase (LCP) crystallization. The purified GFP free NhaA (36 mg mL$^{-1}$) was mixed with molten monoolein (Sigma, CAS No. 111-03-5) in a weight ratio of 2:3, respectively, using coupled syringe-mixing device (Hamilton). A transparent cubic phase was formed and crystallization trials were set up by dispensing 50 nL cubic phase onto a 96-well Laminex glass plate (MD11–50, Molecular Dimensions), which was then covered with 800 nL of crystallization solution (0.1 M MES pH 6.5, 0.1 M NaCl, 0.1 M CaCl$_2$, 24–45% (v/v) PEG 400), with a LCP Mosquito robot (TTP Labtech). Plates were sealed with a Laminex glass cover (MD11–52, Molecular Dimensions) and were stored at 20 °C. The LCP crystallization for Z. mays CMP–sialic acid transporter was carried out as described for E. coli NhaA except that the protein was incubated with 1.2 mM CMP prior to screening the MemMeso™ LCP screen kit (Molecular Dimensions). For the Z. mays CMP–Sialic acid transporter, no detectable LCP crystal were seen without the addition of CMP despite extensive LCP crystallization trials. Likewise, for NhaA, no LCP crystals was obtained when it was purified in the absence of cardiolipin (18:1) in the purification buffers. The LCP crystals obtained were harvested without any cryo-protectant and cryocooled in liquid nitrogen. The diffraction data was collected at ESRF ID30A-3 on an Eiger X 4M detector with 0.968 Å wavelength X-rays with flux 8.65E11 ph s$^{-1}$ at 1° oscillation and 0.05 s exposure.

## Data availability

Data supporting the findings of this manuscript are available from the corresponding author upon reasonable request.

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

## Acknowledgments

We are grateful to Magnus Claesson for stimulating discussions and critically reading of the manuscript, Mathieu Coincon for ITC measurements and Povilas Uzdavinys with the crystallization of NhaA. LCP crystals were screened at ESRF with excellent assistance from beamline scientists. M.L. is supported by an Ingvar Carlsson Award from the Swedish Foundation for Strategic Research and a Karolinska Faculty-funded Career Position. The Swedish Research Council and the Knut and Alice Wallenberg Foundation funded this work (D.D.). D.D. acknowledges support from EMBO through the Young Investigator Program.

## Author contributions

D.D. designed and supervised the project. Experiments were performed by E.N. and Y.C. E.N., Y.C., M.L., and D.D. discussed the results and commented on the manuscript.

## Additional information

**Competing interests:** The authors declare no competing interests.

