## [Peer Review File · Nature Communications]

Reviewers' comments:

Reviewer #1 (Remarks to the Author):

Nji et al have devised a new high throughput assay for determining the stability of unpurified membrane proteins fused to GFP after detergent solubilisation. This assay has the advantage of not requiring any complex equipment apart from a microfuge and a fluorimeter, and will therefore be widely usable and much faster than using FSEC, which requires equipment dedicated to this one assay. The authors benchmark their new assay against an established hFSEC assay and show that the results are the same. They then use the new assay to study the differences in stability between eukaryotic membrane proteins and prokaryotic membranes. This has been done before by Drew's group, and the new assay confirms the previous data that on the whole, eukaryotic membrane proteins are in general less stable. However, in this manuscript they extend these data by showing that purified prokaryotic membrane proteins tend to have similar stability to when they are initially solubilised, whereas eukaryotic membrane proteins generally lose significant stability upon purification. This has long been suspected in the field, but these are the first data on multiple different proteins that perhaps support this contention. The major finding of the authors is that addition of eukaryotic lipids to the purified protein increased the thermal stability of the proteins. This effect was specific for eukaryotic lipids and was not mimicked by purified synthetic lipids.

The data and methodology will be of great interest to researchers engaged in membrane protein structural biology. The manuscript is reasonably well written, but lacks information in a number of places, which needs to be addressed along with the more fundamental issue with respect to using yeast as an expression system.

I have two major criticisms of the experimental finding that the stability of eukaryotic membrane proteins decreases upon purification and that this can be restored or even increased upon adding back lipid. These need to be addressed through additional experiments.

1. It can be argued that the expression system used, *Saccharomyces cerevisiae* is suboptimal for the production of mammalian membrane proteins, because they lack the specific lipids required for the stabilisation of mammalian membrane proteins. Thus the large drop in stability observed for the eukaryotic membrane proteins upon purification is because of the expression system used, not because eukaryotic membrane proteins per se are being purified. The fact you do not see this for the bacterial membrane proteins could be because they are expressed in a bacterial host and any lipids required for stability are already present and carried through into the purified protein. If bacterial membrane proteins were expressed in human cells, the authors may see a decrease in their stability upon purification, which could be rescued by adding back *E. coli* lipids. It is also possible that if the authors expressed the mammalian membrane proteins in human cells, then upon purification they would not see a decrease in stability. Alternatively, the authors could add an excess of brain lipids to detergent-solubilised *S. cerevisiae* membranes, allow time for equilibration, and then purify. It may be that the result will be identical to that already observed, which would be suggestive that the affinity of specific lipids for eukaryotic membrane proteins may be lower than those for bacterial membrane proteins.

2. The comparison in stability between the purified eukaryotic membrane protein and the detergent-solubilised membrane protein needs to be performed under identical conditions to be valid. However, given that the apparent stability of Glut1, human CMP Sia, NHA2 etc are higher in the purified state than in the non-purified state, suggests one of two things. Firstly, that the detergent concentration or buffer conditions are not identical or, secondly, that the lipids from *S. cerevisiae* are a poor substitute for mammalian lipids (i.e. the point above). The detergent concentration upon solubilisation is typically much higher than after purification, so multiple factors may be involved.

Both of the above points need to be addressed, as they impact on the general conclusions of the

paper.

Other points.

1. The authors have, extremely irritatingly, not numbered the pages or lines. I will do my best to highlight where the issues are, but have not dealt with typos because of the onerousness in defining where they are.
2. Introduction, para 1, line 8. Add references to the methods used for studying lipid membrane protein interactions.
3. Introduction para 2, lines 2-4. This statement is nonsense and the authors know it, as they have used the CPM assay for studying the stability of membrane proteins. Add a more cogent argument for using FSEC based on purification versus using non-purified samples. Highlight the problems of time required for FSEC (give numbers) so that readers can see the advantage of the new strategy in the manuscript.
4. Results and Discussion: Validation section para 1, line 9 onwards. It is essential that the authors discuss parameters that will affect the apparent T_m generated by the hFSEC assay, in particular the amount of membranes used and the amount of detergent and its concentration. The authors must state how they have controlled this throughout their experiments.
5. Throughout the manuscript: ensure that the chain length of the lipids used is defined. 'Cardiolipin' is insufficient. Continually going back to Methods is also not an option.
6. Results and Discussion: Validation section para 3, line 6 and also Fig 1d. It is inappropriate to use the term 'Kd' for the value of 1.6 mM. You are not measuring binding directly, you are measuring the change in fluorescence after heating. It would be better to call this an Effective Concentration giving 50% of the observable effect i.e. an EC50 value.
7. Results and Discussion: Developing heat FSEC section para 2, lines 6-7; I do not understand the statement that 'there was no clear melting transition' as according to the data presented in Figure 2 there clearly is.
8. Throughout the manuscript; the term melting temperature and T_m are used interchangeably. Please use 'apparent T_m ' or 'apparent melting temperature' throughout for consistency.
9. Results and Discussion: Screening for Lipid Stabilisation section para 1, line 2; give references to the statement that lipids are progressively lost on purification.
10. The most used abbreviation for cholesterylhemisuccinate is CHS not CHEMS.
11. Throughout; do not refer to 'crude membranes' but use instead 'detergent-solubilised membrane' or 'unpurified detergent-solubilised membrane proteins' for clarity as appropriate.
12. Conclusions para 5 line 14-15; the authors state that CHS mimics the properties of cholesterol quite well, but in detergent micelles there are non-physiological interactions with the detergent that suggests the stabilisation effect is not just like the theoretical effects of cholesterol (Ref 31).
13. Methods; Preparation of membrane proteins para 1, define how E. coli membranes are prepared.
14. Methods; GFP-TS; it is essential to define the concentration of membranes (in terms of mg of membrane protein per ml) used in the solubilisation. Also define how much of a given membrane protein GFP fusion is required for the assays.
15. Methods; lipid screening; para 1; define the source and catalog number of the lipids used. Describe exactly how the stock solutions are made.
16. Incomplete references 18 and 34
17. All figures; define how many times each experiment was done, whether the data shown are from a representative experiment, how many times the fluorescence measurements were made on the same sample (if appropriate) and the errors associated with the T_m calculation based on the curve fit through the data points. Put in error bars where appropriate.
18. Fig 1; define 'Normalised' in the context of the fluorescence signal. How is it that the normalised signal at 1 mM cardiolipin in panel d is about 5 and yet it is 90 in panel e.
19. Fig1, change Kd to EC50.
20. Fig 3 panel e, define which lipid was used on the graph
21. Table 1; the average T_m is meaningless and the error associated with this number incorrectly calculated. I suggest that it would be better to add another column and include the difference

between each measurement (ΔT_m) and give an average of this (and calculate the error correctly).

22. Supplementary table 1; define the lipid used

23. Discussion; Please discuss the variability in slopes observed in the stability data. How do you take this into account when calculating an apparent T_m , as a shallow slope is highly indicative of multiple conformations compared to a sharp transition, which suggests one major conformation. The latter would be preferable even if the apparent T_m s were identical.

Reviewer #2 (Remarks to the Author):

In this manuscript, Dr. Drew and his colleagues applied the FSEC-based thermal shift assay to screen a panel of lipids. In addition, they also developed a method called GFP-TS to screen lipids similarly but without SEC, which would save a significant amount of time for screening. Dr. Drew has worked on the development of GFP-based membrane expression techniques for more than ten years, and the data shown here looks pretty solid. I enjoyed reading this manuscript pretty much.

Major concerns.

A major concern is the novelty of the methods described in this manuscript. The FSEC-TS (called heat-FSEC in this manuscript) was originally developed by Dr. Gouaux, as they cited (PMID: 22884106), and the screening of lipids by FSEC-TS is already described in their paper.

The novelty of GFP-TS is also not so clear to me. GFP-TS seems to have a clear advantage over FSEC-TS. The use of 96 well plates would save a lot of time, as it does not require a time-consuming SEC process. However, the use of 96 well plates for GFP-fusion proteins is already described in the earlier papers by Dr. Drew (PMID: 18451787, PMID: 15987891, etc).

Therefore, I agree that lipid screening by GFP-TS would be a useful tool for membrane protein biochemistry, but I am not sure whether this manuscript is worth publishing in one of the leading journals, Nature Communications. I think, this may be a border line case. I personally like this manuscript pretty much, but I would like to see the strong defense regarding the novelty of this work in the revised manuscript. For instance, it would be great if the authors can show a successful application of GFP-TS, which leads to the new membrane protein structure determination or identification of a novel lipid-membrane protein interaction with physiological relevance, not just the validating of the method. These additional data would significantly strengthen the significance of their work.

Minor concerns.

1. Page 6. "However, without the size-exclusion step there was no clear melting transition, and the calculated T_m was ~ 11 °C higher than that estimated by hFSEC. we incubated the protein solutions with the short-chain non-ionic detergent octyl- β -D-glucoside (β -OG) prior to heating,"

β -OG is well known as a very harsh detergent. So, some of membrane proteins may not be suitable to see the nice thermal shift either by FSEC-TS or by GFP-TS. Likewise, the thermal shift of some of the oligomeric membrane proteins may not be suitable to see the thermal shift by GFP-TS, as they may show the multiple oligomeric peaks after heating. If the authors mention these possible pitfalls, it would be helpful when readers try GFP-TS in the future.

2. Page 11. "Besides re-enforcing the use of the GFP-TS methodology as a screen to detect specific lipid-protein interactions, our data also indicate that the stabilities of membrane proteins may be much more closely related to their respective native membrane environments than previously thought."

I am so sorry, but I did not understand what this sentence meant clearly. It would be great if the authors can provide more explanation.

3. Page 13, "Importantly, the thermostability measurements made by the resulting GFP-TS in

crude-membranes correlated well with the stability estimates in detergent solution from purified samples using the CPM assay, which monitors unfolding of the protein directly³

The stabilities of hGLUT1 and rGLUT5 seem to be quite different before and after purification (Page 10). The stabilities of hGLUT1 WT and E329Q mutant are also very different before and after purification (Page 10). These indicate that the thermostability assay sometimes requires purification, which is a very time consuming process. The authors should address this concern.

4. "heat FSEC" should be changed to "FSEC-TS" in the throughout manuscript, as it is named as FSEC-TS in the original paper.

5. According to the the Reporting Checklist For Nature Communications Life Sciences Articles, "For small sample sizes (n<5) descriptive statistics are not appropriate, instead plot individual." https://media.nature.com/full/nature-assets/ncomms/authors/ncomms_lifesciences_checklist.pdf I am not whether this manuscript matches this policy. Some of the experiments seem to be repeated twice. The authors can check this issue.

Reviewer #3 (Remarks to the Author):

The manuscript by Nji, Chatzikyriakidou, and Drew describe the use of heat FSEC, an established technique (ref 5), to monitor thermal stability and how the presence of detergent solubilized lipids impact thermal stability. The authors also describe the use of an ultra centrifugation based assay and CPM thermal shift assay.

The considerably problem I have with this paper is that it does not present anything new. In fact, ref 5 has examined how different lipids can stabilize a membrane protein. I appreciate the authors have examined a small number of prokaryotic and eukaryotic membrane proteins using heat hFSEC, which differ from ref 5. The authors use the cardiolipin-dependent stabilization of NhaA, to demonstrate specificity of lipid-protein interactions, however these observations have been previously published by the group (ref 6-7). This paper demonstrates the utility of the hFSEC method and more suitable for another journal, such as Anal. Chem. or specialized journal.

The authors state throughout the manuscript that hFSEC can be used to identify specific-lipid protein interactions. Unfortunately, the technique used by authors does not allow quantification of specific-lipid interactions. Native mass spec studies (which are poorly cited in this paper) have demonstrated a number of lipids can bind to a detergent solubilized membrane protein i.e. at a given lipid concentration there will be a distribution of bound lipids. Given this fact, the authors have made the assumption of one lipid binding site per membrane protein, which is clearly not the case. The authors go as far to report a binding affinity constant (for example Fig 1d), which is not even described in methods. It is unclear how the authors can separate the multiple lipid binding events taking place in solution from the hSEC data. In addition, equilibrium binding constants and thermodynamics has been reported for AmtB (DOI: 10.1021/jacs.6b01771 and not cited in this work). How does their binding constant compare, since both are done in detergent? The authors should rather consider reporting an effective concentration (EC50) as this value represents the ensemble of lipid binding events. Importantly, does the curve in Fig 1d change if a different temperature was used?

Another concern is lack of experimental repeats and statistical analysis. No error bars are presented, yet error bars are reported in Table S1. This needs to be corrected throughout. Values are presented with and without error throughout the manuscript. I managed to find buried in methods $n = 2$ but unclear if applies to all the data presented. The authors state claims of significance without rigor. For example, the authors compare Human Glut1 and variant E329Q followed by stating "the conformational constrained E329Q variant is less sensitive to lipid loss". The values in comparison are 19.8 ± 7.8 and 28.5 ± 0.7 . Given error, there is no statistical

difference here, especially for $n=2$.

In the abstract, the authors state "a large cohort of many medically". The small number of membrane protein system studies does not support this claim. Moreover, the data presented does not support the last sentence of the abstract.

The literature is poorly cited and missing many key papers in the field. As an example, in the introduction the authors mention different techniques but provide no references.

An engineered thermal shift screen reveals specific lipid preferences of eukaryotic and prokaryotic membrane proteins

Corresponding authors:

David Drew

We thank the referees for their considered evaluation. We have responded, as appropriate, to all queries below.

Reviewer #1 (Remarks to the Author):

Nji et al have devised a new high throughput assay for determining the stability of unpurified membrane proteins fused to GFP after detergent solubilisation. This assay has the advantage of not requiring any complex equipment apart from a microfuge and a fluorimeter, and will therefore be widely usable and much faster than using FSEC, which requires equipment dedicated to this one assay. The authors benchmark their new assay against an established hFSEC assay and show that the results are the same. They then use the new assay to study the differences in stability between eukaryotic membrane proteins and prokaryotic membranes. This has been done before by Drew's group, and the new assay confirms the previous data that on the whole, eukaryotic membrane proteins are in general less stable. However, in this manuscript they extend these data by showing that purified prokaryotic membrane proteins tend to have similar stability to when they are initially solubilised, whereas eukaryotic membrane proteins generally lose significant stability upon purification. This has long been suspected in the field, but these are the first data on multiple different proteins that perhaps support this contention. The major finding of the authors is that addition of eukaryotic lipids to the purified protein increased the thermal stability of the proteins. This effect was specific for eukaryotic lipids and was not mimicked by purified synthetic lipids.

The data and methodology will be of great interest to researchers engaged in membrane protein structural biology. The manuscript is reasonably well written, but lacks information in a number of places, which needs to be addressed along with the more fundamental issue with respect to using yeast as an expression system.

I have two major criticisms of the experimental finding that the stability of eukaryotic membrane proteins decreases upon purification and that this can be restored or even increased upon adding back lipid. These need to be addressed through additional experiments.

1. It can be argued that the expression system used, *Saccharomyces cerevisiae* is suboptimal for the production of mammalian membrane proteins, because they lack the specific lipids required for the stabilization of mammalian membrane proteins. Thus the large drop in stability observed for the eukaryotic membrane proteins upon purification is because of the expression system used, not because eukaryotic membrane proteins per se are being purified. The fact you do not see this for the bacterial membrane proteins could be because they are expressed in a bacterial host and any lipids required for stability are already present and carried through into the purified protein. If bacterial membrane proteins were expressed in human cells, the

authors may see a decrease in their stability upon purification, which could be rescued by adding back *E. coli* lipids. It is also possible that if the authors expressed the mammalian membrane proteins in human cells, then upon purification they would not see a decrease in stability. Alternatively, the authors could add an excess of brain lipids to detergent-solubilised *S. cerevisiae* membranes, allow time for equilibration, and then purify. It may be that the result will be identical to that already observed, which would be suggestive that the affinity of specific lipids for eukaryotic membrane proteins may be lower than those for bacterial membrane proteins.

Thank for raising this important consideration. Our data is consistent with many observations that purified eukaryotic membrane proteins are less stable than purified bacterial membrane proteins (e.g., *Structure*. 2011 Jan 12; 19(1): 17–25.). The surprise findings here are that the eukaryotic membrane proteins melting temperatures prior to purification are almost as good as the bacterial ones. This, in itself, already indicates that the eukaryotic membrane proteins are well-folded in yeast and we have already ruled out there was nothing special about the *E. coli* lipids, since their addition did not help the stability of the purified eukaryotic membrane proteins.

Nonetheless, to strengthen our conclusions we have carried out the following additional experiments:

1. We combined equal amounts of detergent solubilized *E. coli* and yeast membranes to the purification of Xyle and rat GLUT5 (**Methods, Pg 20, line 813**). We saw no increase in either of the melting temperatures of Xyle and GLUT5 prior to purification and again GLUT5 was significantly more unstable after purification. Essentially, the results are identical to before. (**Manuscript, Pg 10, line 343 and Supplementary Fig. 4b**).
2. We isolated membranes from human kidney embryonic cells (HEK293) and added equal amounts of the detergent solubilized membranes to human NHA2 (**Methods, Pg 20, line 821**). We saw no improvement in the stability prior to purification and again we saw the same drop in stability after purification as when they were not added (**Manuscript, Pg 11, line 378 and Supplementry Fig. 4b**).

We kindly thank the referee for suggesting these experiments, which we feel have helped strengthened the conclusions reached in the initial submission.

The comparison in stability between the purified eukaryotic membrane protein and the detergent-solubilised membrane protein needs to be performed under identical conditions to be valid. However, given that the apparent stability of Glut1, human CMP Sia, NHA2 etc are higher in the purified state than in the non-purified state, suggests one of two things. Firstly, that the detergent concentration or buffer conditions are not identical or, secondly, that the lipids from *S. cerevisiae* are a poor substitute for mammalian lipids (i.e. the point above). The detergent concentration upon solubilisation is typically much higher than after purification, so multiple factors may be involved.

Thank you for this to point. To clarify, we think the reviewer is asking if the stability of the eukaryotic membrane proteins is higher in the **unpurified** state than in the **purified** state (and not the other way as written), because of the difference in the percentage of detergent used between the unpurified material vs. the purified fusions?

1% (w/v) of DDM is solubilizing 3.5 mg/ml worth of **total** protein. The amount of GFP-fusion in this fraction is less than 1% of this amount. It is reasonable to assume we are measuring unfolding in the presence of saturating levels of lipid and detergent. Once purified, our fusion concentration represents the total levels. We have carried out the assays at 5,000 RFU levels, which corresponds to a protein concentration of ~ 0.04 mg/ml. To keep a similar amount of detergent for 0.04 mg/ml of protein we should have at least 0.01% DDM. Since this is close to the CMC of DDM to be on the safe side we have used 0.03% DDM in our assays. As we agree this is important to clarify we have added the solubilization details the main text (**Pg 9, line 306**) and also more detailed Methods (**Pg 18, line 723**).

In addition to above, the best argument that the higher amount of detergent used during solubilization has not had a biased stabilizing effect is that this should have stabilized the bacterial membrane proteins under the identical conditions. However, this is clearly not the case, since we only saw a drop in the stability for the purified eukaryotic membrane proteins, but not the bacterial membrane proteins (**Fig. 3a**).

Both of the above points need to be addressed, as they impact on the general conclusions of the paper.

Other points.

1. The authors have, extremely irritatingly, not numbered the pages or lines. I will do my best to highlight where the issues are, but have not dealt with typos because of the onerousness in defining where they are.

Our apologies. We have now numbered the lines and pages.

2. Introduction, para 1, line 8. Add references to the methods used for studying lipid membrane protein interactions.

We apologize for not properly referencing the field and it was an unintentional oversight. We have now added modified the introduction (main text pg 3, line 56) and added the appropriate references.

3. Introduction para 2, lines 2-4. This statement is nonsense and the authors know it, as they have used the CPM assay for studying the stability of membrane proteins. Add a more cogent argument for using FSEC based on purification versus using non-purified samples. Highlight the problems of time required for FSEC (give numbers) so that readers can see the advantage of the new strategy in the manuscript.

Your'e right, in hindsight this sentence was too ambiguous. We were referring to the thermofluor dyes that bind through hydrophobic interactions, but we should have been clearer that the dyes that use conjugation to free thiol groups, such as the CPM

assay, are still compatible. We have re-written the sentence and listed the advantages of FSEC-TS compared to the CPM assay (**main text pg 3, line 72**).

In brief, the main advantage relevant to this study is that the CPM assay requires purified protein. There are further advantages too. The main ones are that the CPM assay uses more protein than FSEC-TS and it can sometimes give uninterpretable results. For globular proteins its estimated that 1/3 of proteins are incompatible with the CPM assay and Robert Strouds group estimated these levels are higher for membrane proteins (Tomasiak, T. M. et al. *Curr Protoc Protein Sci* 77, 29 11 21-14, 2014). In our experience, it can be difficult to accurately measure the unfolding of unstable membrane proteins, which tend to be those from eukaryotic origin. We interpret the main problem is that the CPM dye cannot distinguish between a misfolded aggregate versus a folded protein. It was for this reason we used the assay to calculate relative unfolding times at a constant temperature of 40°C when comparing the purified stability of bacterial versus eukaryotic membrane proteins (*Structure*. 2011 Jan 12;19(1):17-25). We also found we couldn't use β -OG as aggregated proteins precipitated in this detergent and gave unreliable results. Even mild detergent like DDM can be problematic, which is why we think stability measurements made in LDAO are the most reliable since this detergent is good at keeping aggregates in solution. Obviously, harsher detergent are not ideal, however, for monitoring lipid-protein interactions.

Thank you for the suggestion that we should highlight the time-saving advantages of the modified assay (GFP-TS) *versus* FSEC-TS. The approximate time to measure an FSEC-TS titration was 3-4 hours (depending on the column) and 1 min for GFP-TS. This information has been added to the Methods section of the manuscript (**methods pg 19, line 767**) and touched upon in the discussion (**main text pg 14, line 512**). Furthermore, unlike FSEC-TS, the GFP-TS method can be easily parallelized to increase the number of samples with little impact on handling time, *i.e.*, FSEC-TS titrations on 3 different proteins in duplicate will take several days to run and require an automated sampler.

4. Results and Discussion: Validation section para 1, line 9 onwards. It is essential that the authors discuss parameters that will affect the apparent T_m generated by the hFSEC assay, in particular the amount of membranes used and the amount of detergent and its concentration. The authors must state how they have controlled this throughout their experiments.

Thank you raising this point and we should have been clearer. We aimed keep to a total final protein concentration of 3.5 mg/ml with 1% DDM, which we have now added to both the methods and main text (**pg 9, line 306; Pg 19, line 748**).

For two of the low expressing eukaryotic membrane proteins we have to have a total final protein concentration of 6 mg/ml so that the fluorescence signal of the extracted material would be high enough to measure. To ensure that this difference would not influence the apparent T_m we compared the melting temperatures of ASBT over a range of total protein concentrations in 1% DDM. This control has been added to the main text (**pg 9, line 310**) and **Supplementary Fig. 1d**.

5. Throughout the manuscript: ensure that the chain length of the lipids used is

defined. 'Cardiolipin' is insufficient. Continually going back to Methods is also not an option.

OK. This has now been updated throughout the main text and methods.

6. Results and Discussion: Validation section para 3, line 6 and also Fig 1d. It is inappropriate to use the term 'Kd' for the value of 1.6 mM. You are not measuring binding directly, you are measuring the change in fluorescence after heating. It would be better to call this an Effective Concentration giving 50% of the observable effect i.e. an EC50 value.

We have thought about this suggestion in detail. Our understanding is that an EC50 refers to a concentration that results in 50% drop in biological activity, *i.e.*, of an enzyme reaction. However, in this assay we are not measuring the effect on biological activity, but the strength of the interaction. Quantitative methods can determine dissociation constants (K_d) from the fraction of complex formed as a function of ligand concentrations from changes in heat (Protein Sci. 2011 Aug; 20(8): 1439–1450.). The appropriate use is in this case is a dissociation constants (K_d) and we have clarified that we based on the dimeric crystal structure of NhaA we assume a single binding site for cardiolipin (**pg 5, line 153**).

7. Results and Discussion: Developing heat FSEC section para 2, lines 6-7; I do not understand the statement that 'there was no clear melting transition' as according to the data presented in Figure 2 there clearly is.

Thank you. We have modified the sentence to better reflect the data.

8. Throughout the manuscript; the term melting temperature and T_m are used interchangeably. Please use 'apparent T_m ' or 'apparent melting temperature' throughout for consistency.

OK. We can confirm this has now been corrected throughout the manuscript.

9. Results and Discussion: Screening for Lipid Stabilisation section para 1, line 2; give references to the statement that lipids are progressively lost on purification.

OK. We can confirm that we have added several references for this statement as highlighted in yellow in the reference section.

10. The most used abbreviation for cholesterylhemisuccinate is CHS not CHEMS.

OK. We can confirm this has now been corrected throughout the manuscript.

11. Throughout; do not refer to 'crude membranes' but use instead 'detergent-solubilised membrane' or 'unpurified detergent-solubilised membrane proteins' for clarity as appropriate.

OK. We can confirm this has now been corrected throughout the manuscript.

12. Conclusions para 5 line 14-15; the authors state that CHS mimics the properties of cholesterol quite well, but in detergent micelles there are non-physiological interactions with the detergent that suggests the stabilisation effect is not just like the theoretical effects of cholesterol (Ref 31).

We agree that in detergent micelles there are non-physiological interactions with CHS. However, Ref 31. also concluded that they stabilization seen by CHS was consistent to that seen by cholesterol “*simulations of A2AR-GL31 in DDM supplemented with CHS showed an appreciable increase in α -helicity and a decrease in RMSD compared to the crystallized receptor after MD simulations (Fig. 6A-D); this effect was also observed in simulations of the receptors in a POPC lipid bilayer with cholesterol*”. In this paper MD simulations of CHS to A2AR was found to form similar interactions as cholesterol in the crystal structure of β 2-adrenergic receptor (PDB code 3D4S), Fig. 6. Indeed, like many other mammalian proteins, it is not possible to purify A2AR unless CHS is added during purification (Eur J Biochem. 2002 Jan; 269(1):82-92). However, the thermostabilized receptor no longer requires the addition of CHS (Neuropharmacology. 2011 Jan; 60(1):36-44), which is consistent with the reasoning that conformational stabilized proteins are less dependent on lipids to restrict their dynamics in detergent, which also contributed to their purified stability.

In light of your comment we have modified the sentence to be more concise (**pg 16, line 587**).

13. Methods; Preparation of membrane proteins para 1, define how E. coli membranes are prepared.

OK. We can confirm that this has now been updated (**Methods pg 17, line 655**).

14. Methods; GFP-TS; it is essential to define the concentration of membranes (in terms of mg of membrane protein per ml) used in the solubilisation. Also define how much of a given membrane protein GFP fusion is required for the assays.

OK. We can confirm that this has now been updated (**Methods, pg 19, line 754**).

15. Methods; lipid screening; para 1; define the source and catalog number of the lipids used. Describe exactly how the stock solutions are made.

OK. We can confirm that this has now been updated (**Methods, pg 21, line 855**).

16. Incomplete references 18 and 34

Thank you. We can confirm that these references have now been updated.

17. All figures; define how many times each experiment was done, whether the data shown are from a representative experiment, how many times the fluorescence measurements were made on the same sample (if appropriate) and the errors associated with the T_m calculation based on the curve fit though the data points. Put in error bars where appropriate.

We apologize this was not clearer. We have made it much clearer in the Methods and Figure legends that the tabulated T_m values are the average from 2 independent solubilization and purifications. The raw data from each one of these melting curves are shown in either the main figures or supplementary figures. Each apparent T_m was calculated from fitting the fluorescence across 9 different temperatures measured, which was each measured in duplicate (18 data points). We have now shown that the robustness of fit (R^2) for each titration was, in almost all, cases 0.98-0.99. Each temperature was an average of two technical repeats and we now include error-bars that show the range of two data points (**Methods pg 19, line 768**). We made the decision to only use two technical repeats as we found it an unnecessary use of precious samples to do more than two, since the curve-fitting was already excellent and the variance from more technical repeats ($n = 6$), was smaller than the fitting error associated with each T_m estimation (see figure below). Indeed, if T_m values are calculated from only one measurement at each temperature the T_m values are, with one exception (X3), within then mean \pm s.e.m. of the fit calculated from the average of 6 technical repeats. If 2 technical repeats are used the calculated T_m is always within error of the calculation made from 6 technical repeats.

X1 T_m	X2 T_m	X3 T_m	X4 T_m	X5 T_m	X6 T_m	Average X1-X6
40.42 \pm 0.28	40.77 \pm 0.36	39.53 \pm 0.51	40.20 \pm 0.38	40.32 \pm 0.47	40.39 \pm 0.30	40.31 \pm0.16

18. Fig 1; define 'Normalised' in the context of the fluorescence signal. How is it that the normalised signal at 1 mM cardiopin in panel d is about 5 and yet it is 90 in panel e.

Fig.1d has been normalized across the **whole** series with 100% set as the highest value. This data does not need to be normalized, but for sake of consistency we have converted the fluorescence counts into an arbitrary range from 0 to 100. In **Fig. 1e.**, however, because of the large difference in fluorescence amplitudes each curve was

separately normalized (re-scaled) to make it easier to compare elution volumes of the respective peaks.

19. Fig1, change K_d to EC50.

With all due respect we have clarified the use of K_d as we think this is more appropriate than EC50.

20. Fig 3 panel e, define which lipid was used on the graph

OK. We can confirm that this has now been updated in the manuscript.

21. Table 1; the average T_m is meaningless and the error associated with this number incorrectly calculated. I suggest that it would be better to add another column and include the difference between each measurement (ΔT_m) and give an average of this (and calculate the error correctly).

Point taken. Thank you for this suggestion. In **Table 1**, the last column shows the T_m difference estimates between the two methods. We have not included an error of this difference as we think it is clear from the fitting errors that the differences between the two methods are within a similar range, as explained above.

22. Supplementary table 1; define the lipid used

OK. We can confirm that this has now been updated to **Supplementary Table 1**.

23. Discussion; Please discuss the variability in slopes observed in the stability data. How do you take this into account when calculating an apparent T_m , as a shallow slope is highly indicative of multiple conformations compared to a sharp transition, which suggests one major conformation. The latter would be preferable even if the apparent T_m s were identical.

We assume a single melting transition in the calculation of a T_m . Although there are differences in the slope of the melting transition and the referee raises an intriguing point, we do not think the thermal-based denaturation methods are sensitive enough to dissect these differences and therefore we have not discussed them further.

Reviewer #2 (Remarks to the Author):

In this manuscript, Dr. Drew and his colleagues applied the FSEC-based thermal shift assay to screen a panel of lipids. In addition, they also developed a method called GFP-TS to screen lipids similarly but without SEC, which would save a significant amount of time for screening. Dr. Drew has worked on the development of GFP-based membrane expression techniques for more than ten years, and the data shown here looks pretty solid. I enjoyed reading this manuscript pretty much.

Major concerns.

A major concern is the novelty of the methods described in this manuscript. The FSEC-TS (called heat-FSEC in this manuscript) was originally developed by Dr.

Gouaux, as they cited (PMID: 22884106), and the screening of lipids by FSEC-TS is already described in their paper.

The novelty of GFP-TS is also not so clear to me. GFP-TS seems to have a clear advantage over FSEC-TS. The use of 96 well plates would save a lot of time, as it does not require a time-consuming SEC process. However, the use of 96 well plates for GFP-fusion proteins is already described in the earlier papers by Dr. Drew (PMID: 18451787, PMID: 15987891, etc).

Therefore, I agree that lipid screening by GFP-TS would be a useful tool for membrane protein biochemistry, but I am not sure whether this manuscript is worth publishing in one of the leading journals, Nature Communications. I think, this may be a border line case. I personally like this manuscript pretty much, but I would like to see the strong defense regarding the novelty of this work in the revised manuscript. For instance, it would be great if the authors can show a successful application of GFP-TS, which leads to the new membrane protein structure determination or identification of a novel lipid-membrane protein interaction with physiological relevance, not just the validating of the method. These additional data would significantly strength the significance of their work.

Thank you for your comments. In the manuscript, we have put the emphasis of the methodology to help dissect a general important physiological question. Using the GFP-TS assay, we are able to address the key role of lipids and their differences in the stabilization and oligomerization of bacterial vs. eukaryotic membrane proteins (**Fig 3a**). However, we appreciate the reviewer's suggestion to include a concrete example demonstrating a GFP-TS-enabled discovery. In light of this consideration, we have now used the GFP-TS assay to uncover the destabilization caused by monoolein, a lipid used in lipid-cubic phase (LCP) crystallization, to some of the eukaryotic membrane proteins (**main text pg 12, line 438; Supplementary Fig. 6a**).

To see if the destabilizing effects of monoolein could be compensated for by the addition of a stabilizing lipid, we repeated the analysis of NhaA with and without the addition of cardiolipin. Indeed, the apparent T_m of NhaA was significantly decreased in the presence of monoolein only (from 42 to at 27 °C), but rescued in the presence of cardiolipin, (41°C), **Supplementary Fig. 6b, c**. As proof-of-principle of this approach, we could obtain LCP crystals of NhaA when supplemented with cardiolipin and further improve the published structural resolution from 3.5 Å to 2.3 Å, **Supplementary Fig. 6d, e**.

Furthermore, we have had success applying this approach to eukaryotic membrane protein structure determination too, but we maintain the inclusion of these structures goes beyond the current scope of the paper. With all due respect, we also do not want the important biology to be lost by the addition of too much methods-based examples.

Minor concerns.

1. Page 6. "However, without the size-exclusion step there was no clear melting transition, and the calculated T_m was ~11 °C higher than that estimated by hFSEC. we incubated the protein solutions with the short-chain non-ionic detergent octyl- β -D-glucoside (β -OG) prior to heating,"

β -OG is well known as a very harsh detergent. So, some of membrane proteins may not be suitable to see the nice thermal shift either by FSEC-TS or by GFP-TS. Likewise, the thermal shift of some of the oligomeric membrane proteins may not be suitable to see the thermal shift by GFP-TS, as they may show the multiple oligomeric peaks after heating. If the authors mention these possible pitfalls, it would be helpful when readers try GFP-TS in the future.

Thank you for pointing this out. Indeed, when we tested using only β -OG we could not measure the melting temperature for some of the more unstable membrane proteins (data not shown). For this reason, all the membrane proteins are first solubilized in the mild detergent DDM. β -OG is only added at the last-step prior to heating and denaturation. Its inclusion helps to precipitate any of the aggregation caused by heating. Comparing the apparent T_m in **Supplementary Fig. 1b** (minus β -OG addition) to **Table 1** (plus β -OG addition) once can see that β -OG only lowers the T_m , on average, by 5°C. As such, we think this method should be compatible for most membrane proteins.

We agree that it is possible that due to oligomerization or complex formation that the assay may overestimate the T_m . We have added a caveat to the discussion that the GFP-TS parameters might need to be adjusted in some cases to match FSEC-TS before carrying out HTP ligand-binding assays in particular (**pg 14, line 517**).

2. Page 11. “Besides re-enforcing the use of the GFP-TS methodology as a screen to detect specific lipid-protein interactions, our data also indicate that the stabilities of membrane proteins may be much more closely related to their respective native membrane environments than previously thought.”

I am so sorry, but I did not understand what this sentence meant clearly. It would be great if the authors can provide more explanation.

We agree this sentence is too ambiguous. It was meant to reflect the findings that the eukaryotic membranes proteins are not (necessarily) more unstable than the bacterial ones in a lipid-rich environment, which we have monitored using the GFP-TS assay in the unpurified state (Fig. 3a). We have now removed the sentence and saved these statements for the Discussion, (**pg 15, line 554**).

3. Page 13, “Importantly, the thermostability measurements made by the resulting GFP-TS in crude-membranes correlated well with the stability estimates in detergent solution from purified samples using the CPM assay, which monitors unfolding of the protein directly”

The stabilities of hGLUT1 and rGLUT5 seem to be quite different before and after purification (Page 10). The stabilities of hGLUT1 WT and E329Q mutant are also very different before and after purification (Page 10). These indicate that the thermostability assay sometimes requires purification, which is a very time consuming process. The authors should address this concern.

The assay can work on unpurified or purified material and we used this fact to analyze the stability of membrane proteins before and after purification. Rather, than a concern, this is an interesting finding as it suggests that the eukaryotic membrane

proteins are more sensitive to lipid loss than the bacterial membranes proteins, which we think is correlated with the fact they have evolved to function in more complex membrane environments than the bacterial proteins.

4. "heat FSEC" should be changed to "FSEC-TS" in the throughout manuscript, as it is named as FSEC-TS in the original paper.

OK. We can confirm that this has now been updated throughout the manuscript.

5. According to the the Reporting Checklist For Nature Communications Life Sciences Articles, "For small sample sizes (n<5) descriptive statistics are not appropriate, instead plot individual." https://media.nature.com/full/nature-assets/ncomms/authors/ncomms_lifesciences_checklist.pdf

I am not whether this manuscript matches this policy. Some of the experiments seem to be repeated twice. The authors can check this issue.

Thank you. Please see response to Referee 1 Q17.

Reviewer #3 (Remarks to the Author):

The manuscript by Nji, Chatzikyriakidou, and Drew describe the use of heat FSEC, an established technique (ref 5), to monitor thermal stability and how the presence of detergent solubilized lipids impact thermal stability. The authors also describe the use of an ultra centrifugation based assay and CPM thermal shift assay.

The considerably problem I have with this paper is that it does not present anything new. In fact, ref 5 has examined how different lipids can stabilize a membrane protein. I appreciate the authors have examined a small number of prokaryotic and eukaryotic membrane proteins using heat hFSEC, which differ from ref 5. The authors use the cardiolipin-dependent stabilization of NhaA, to demonstrate specificity of lipid-protein interactions, however these observations have been previously published by the group (ref 6-7). This paper demonstrates the utility of the hFSEC method and more suitable for another journal, such as Anal. Chem. or specialized journal.

Reference 5 is the FSEC-TS assay (Structure. 2012 Aug 8;20(8):1293-9). This paper showed that one could monitor the effects of adding lipids to membrane proteins but did not validate the results to show that it was detecting specific lipid interactions. Indeed, since being published more than 6 years ago, as far as we are aware, no one has published the use of the FSEC-TS assay to monitor lipid-protein interactions.

We have modified FSEC-TS so that is significantly faster and amenable to parallelization (**Please see response to Referee 1, Q3**). We now include data to show how it can further aid structural studies of membrane proteins (**main text pg 12, line 438; Supplementary Fig. 6**).

Arguably the most interesting part of the paper is the novel ability of the GFP-TS assay to directly compare the stabilities of unpurified eukaryotic and bacterial membrane proteins and thus enable such measurements in previously inaccessible biological contexts (**Fig. 3**). We think the field will find the results are unexpected and

surprising. Clearly, as demonstrated by NhaA, this methodology nicely complements other methods such as native MS.

The authors state throughout the manuscript that hFSEC can be used to identify specific-lipid protein interactions. Unfortunately, the technique used by authors does not allow quantification of specific-lipid interactions. Native mass spec studies (which are poorly cited in this paper) have demonstrated a number of lipids can bind to a detergent solubilized membrane protein i.e. at a given lipid concentration there will be a distribution of bound lipids. Given this fact, the authors have made the assumption of one lipid binding site per membrane protein, which is clearly not the case. The authors go as far to report a binding affinity constant (for example Fig 1d), which is not even described in methods. It is unclear how the authors can separate the multiple lipid binding events taking place in solution from the hSEC data. In addition, equilibrium binding constants and thermodynamics has been reported for AmtB (DOI: 10.1021/jacs.6b01771 and not cited in this work). How does their binding constant compare, since both are done in detergent? The authors should rather consider reporting an effective concentration (EC50) as this value represents the ensemble of lipid binding events. Importantly, does the curve in Fig 1d change if a different temperature was used?

The reviewer raises the valid point that native MS is a sensitive method to identify stabilizing lipid interactions with membrane proteins. However, GFP-TS is also able to detect lipid-mediated stabilization in crude extracts and complex lipid mixtures where native MS analysis would be challenging due to the heterogeneity of the interactions. Given that ion mobility MS can be used to uncover allosteric stabilization in simple lipid mixtures (Patrick et al, PNAS 115, 2976-2981, 2018), our study raises the interesting possibility of combining the robust GFP-based stability assay with more sensitive MS-based stability measurements to reveal for example whether a stabilizing effect is imparted by interactions with a specific lipid, or by changing general properties of the lipid/detergent environment.

The reviewer is also right to point out that lipids can attach to a large number of sites on a membrane protein. MS shows that the number of bound lipids will increase nearly linearly with lipid concentration (Barrera et al, Nat Methods 6, 585-587, 2009; Bechara et al, Nat Chem 7, 255-262, 2014; Marty et al, Angew Chem Int Ed 55, 550-554, 2015). Since lipids will associate with detergent-solubilized proteins in a relatively non-discriminating manner (Landreh et al, Anal Chem 89, 7425-7430, 2017), they are likely to form both stabilizing as well as non-stabilizing interactions (Landreh et al, Curr Opin Struct Biol 39, 54-60, 2016). The contributions from the fraction of bound lipids that provides structurally important interactions can be studied by IM-MS (Laganowsky et al, Nature 510, 172-175, 2015). In line with the findings from Laganowsky et al, we reason that the number of sites with structural importance is limited, as evident from the identification of a well-defined binding site for CDL on NhaA (Gupta et al, Nature 541, 421-424, 2017). We assume that only lipids at this site contribute significantly to NhaA stability, in good agreement with the saturation of NhaA stabilization by Cardiolipin observed in the present study. Indeed, we observed non-protein electron density in the dimeric structure maps of NhaA. This was modelled as DDM and two sulfate anions (J Gen Physiol. 144, 529-544, 2014). However, this electron density could also accommodate a single cardiolipin molecule. Indeed, the DDM and sulfate anions are positioned between two clusters of arginine

residues. Arginine residues have shown to be hot-spots for cardiolipin binding (Planas-Iglesias, J et al. Biophysics J. 109, 1282-1294, 2015). We have now better incorporated our analysis of the dimeric NhaA crystal structure into the manuscript (main text, pg 5, line 153).

The reviewer also raises the interesting point to compare binding constants from GFP-based stability assays and native MS (Cong et al, JACS 138, 4346-4349, 2016). However, to be able to compare lipid affinities, the same detergents should be used, since they likely compete with lipids for binding

Laslty, does the K_d change if different temperatures are used? The temperature influences the signal to noise ratio, which if done incorrectly will influence the reliability of the affinity measurement. Under the conditions we used, we see a very sharp binding transition for cardiolipin binding, which means we have a good signal-to-noise. These conditions were optimized based on separate experiments in our group to measure ligand-binding to the plant nucleotide-sugar transporter, i.e., under these conditions the ITC and GFP-TS measurements were very consistent.

CMP binding to the plant CMP-sialic acid transporter was compared by ITC (isothermal titration calorimetry) and also by the GFP-TS assay. In the GFP-TS assay increasing concentrations of CMP were added at a temperature 5 °C higher than the apparent T_m . As such, these parameters were also used to assess cardiolipin binding to NhaA.

Another concern is lack of experimental repeats and statistical analysis. No error bars are presented, yet error bars are reported in Table S1. This needs to be corrected throughout. Values are presented with and without error throughout the manuscript. I managed to find buried in methods $n = 2$ but unclear if applies to all the data presented.

Thank you. Please see response to Referee 1 Q17.

The authors state claims of significance without rigor. For example, the authors compare Human Glut1 and variant E329Q followed by stating “the conformational constrained E329Q variant is less sensitive to lipid loss”. The values in comparison are 19.8 +/- 7.8 and 28.5 +/- 0.7. Given error, there is no statistical difference here, especially for n=2.

Thank you pointing this out. GLUT1 becomes very unstable during purification. Because of this we had a larger-than-average divergence between the two independent purifications. Although this did not affect the conclusion drawn from the comparisons to the bacterial proteins, in hindsight it should have been repeated to make any conclusions regarding the reality stability to the E329Q variant.

As such, we repeated the GLUT1 purification twice more and also in parallel with another purification of the E329Q variant. We now have updated the purified averaged apparent Tm for GLUT1, which is now 27°C (**main text pg 10, line 340**). As such, we conclude that under the parameters tested the purified inward-arrested E329Q mutant is not significantly more stable than the wildtype protein despite it being required for structural determination of human GLUT1 (Nature. 2014 Jun 5;510(7503):121-5).

We find it is interesting that the inward-arrested GLUT1 mutant is not more stable than wildtype GLUT1 in the unpurified state, which indicates that lipids are probably restricting dynamics of the wildtype protein so that it behaves similarly to an inward-locked mutant. Its plausible that lipids are still able to restrict dynamics of wildtype GLUT1 in the purified state too. Indeed, LCP lipids have been seen to interact both to gating helices and the substrate in the homologous protein GLUT3 (Nature. 2015 Oct 15;526(7573):391-6). Indeed, if we continue to purify the mutant GLUT1 vs wildtype GLUT1 (GFP removal and gel filtration steps) we see that the mutant is less prone to aggregation, which indicates wildtype GLUT1 is more unstable and sensitive to delipidation.

Although, we think that this data is interesting and something we are looking into further, we think that the data is too premature at this stage and, as such, have decided not to include the data of the point mutant in the revised version.

In the abstract, the authors state “a large cohort of many medically”. The small number of membrane protein system studies does not support this claim. Moreover, the data presented does not support the last sentence of the abstract. The literature is poorly cited and missing many key papers in the field. As an example, in the introduction the authors mention different techniques but provide no references.

With all due respect, considering the obstacles that have to be overcome during the production of many membrane proteins, and in particular human drug targets, we consider the number of targets in the present study to be comparably large. While some bacterial proteins such as AmtB can of course be produced and purified easily in large amounts, others, such as the eukaryotic GLUT family, human CMP-Sialic acid transporter, and human NHA2, pose significant challenges which vary

considerably between proteins. This has hampered systematic investigations of membrane protein stabilities in purified and unpurified states under comparable conditions, and as a result, previous studies focused on fewer protein systems than the present study (see e.g. Refs 6 and 13, which demonstrated assays using three proteins each). In addition, our dataset includes several homologues, in case of glucose transporters and sodium/proton antiporters even from both bacterial and eukaryotic sources.

Following the referee's suggestion, we have modified the introduction to better reflect the currently available methods to study membrane protein-lipid interactions (**pg 3 line 56**) and have highlighted GFP-TS as a novel approach that nicely complements native MS (**pg 16, line 599**).

Reviewers' comments:

Reviewer #1 (Remarks to the Author):

The authors have dealt with the majority of the suggestions from the reviewers and it now reads very well. Given the potential benefits of the GFP-TS assay for improving the throughput of assays of membrane protein stability and the breadth of data presented, I think it is a great contribution to the field. However, there are still a few issues of clarification the authors need to make.

1. The authors still have not clarified the data shown on graphs and the errors involved. For example in Fig 1a is the data shown from a single representative experiment with a single measurement per data point (no error bars expected) or from two independent experiments with a single measurement per data point (error bars expected) but the error bars are smaller than the symbols. The authors must go through each graph or group of graphs and make this clear. It is fine to say at the end of e.g. the eight graphs in Sup Fig 3 that "all graphs depict data from a single representative experiment with each data point determined once" or whatever the data actually do represent.

2. When two referees independently suggest that K_d is an inappropriate term to be used in the context of an experiment, I would have thought that the authors might take notice. It is, of course, the prerogative of the authors to make fools of themselves if they so wish, but at the end of the day we should all strive for scientific accuracy. Please compare the shape of the curve in Sup Fig 1 and google "shape of a curve of a single-site saturation binding assay".

Reviewer #2 (Remarks to the Author):

I carefully read the revised manuscript. I think, basically the authors addressed most of my concerns, but I still have two minor concerns as follows before publication of this manuscript.

1. Last time I requested the authors to show a successful application of GFP-TS, which leads to the new membrane protein structure determination or identification of a novel lipid-membrane protein interaction with physiological relevance, not just the validating of the method. While the authors showed the LCP crystallization trials of NhaA in the revised manuscript, I do not think it fully meets my previous request. To properly address my concern, I would like the authors to add the following descriptions and data.

a. The detail of the LCP crystallization is not clear. Did the space group and cell dimension change after addition of cardiolipin? Did the authors see cardiolipin in the LCP structure crystallized in the presence of cardiolipin? I understand that the authors do not want to hurt the novelty of the NhaA LCP structure. So, it is not necessary to describe the details of the cardiolipin binding site, but they should answer the above two questions.

b. Similarly, I understand that they do not want to show the names of proteins, whose unpublished structures were determined based on GFP-TS. However, they should describe the results without showing the protein names. For instance, they can say "Protein X, without addition of lipid Y identified by GFP-TS, 7Å resolution. With addition of lipid Y, 2.5Å resolution (Diffraction images)". They can make this kind of a short list and figure without hurting the novelty of their unpublished structures.

2. Another concern is about the comments from Reviewer #3. Reviewer #3 has a serious of concerns, and I basically agreed to the view of Reviewer #3. So, I think for the acceptance of this manuscript, the support from Reviewer #3 would be requires. I would support the publication of this article if Reviewer #3 is happy with the current responses to the reviewer #3.

Reviewer #3 (Remarks to the Author):

The revised manuscript is greatly improved over their original submission. It also includes some new and interesting data. This manuscript nicely adds to a number of key papers pointing to the important role of specific lipid-protein interactions.

Minor comments

I agree with Drew that membrane proteins are difficult to express and purify, the term "large cohort" should be removed. I don't see the point of this and, in my opinion, leaving this in detracts from the paper.

The improved X-ray diffraction of NapA is a nice addition. Could the authors include a figure in main text highlighting the diffraction and electron density maps? I also left wondering if cardiolipin is resolved in higher resolution crystals?

An engineered thermal shift screen reveals specific lipid preferences of eukaryotic and prokaryotic membrane proteins

Corresponding authors:

David Drew

We thank the referees for their considered evaluation. We have responded, as appropriate, to all queries below.

Reviewers' comments:

Reviewer #1 (Remarks to the Author):

The authors have dealt with the majority of the suggestions from the reviewers and it now reads very well. Given the potential benefits of the GFP-TS assay for improving the throughput of assays of membrane protein stability and the breadth of data presented, I think it is a great contribution to the field. However, there are still a few issues of clarification the authors need to make.

1. The authors still have not clarified the data shown on graphs and the errors involved. For example in Fig 1a is the data shown from a single representative experiment with a single measurement per data point (no error bars expected) or from two independent experiments with a single measurement per data point (error bars expected) but the error bars are smaller than the symbols. The authors must go through each graph or group of graphs and make this clear. It is fine to say at the end of e.g. the eight graphs in Sup Fig 3 that "all graphs depict data from a single representative experiment with each data point determined once" or whatever the data actually do represent.

Thank you. We have now clarified the measurements for every experiment. Furthermore, in line with Nature Communications policy each bar graph now shows the individual data points.

2. When two referees independently suggest that K_d is an inappropriate term to be used in the context of an experiment, I would have thought that the authors might take notice. It is, of course, the prerogative of the authors to make fools of themselves if they so wish, but at the end of the day we should all strive for scientific accuracy. Please compare the shape of the curve in Sup Fig 1 and google "shape of a curve of a single-site saturation binding assay".

Thank you, we appreciate the well-placed criticism. We agree that we were mistaken and that a single-binding site K_d is inappropriate in this context. We have decided to refrain from using a parameter to fit the lipid-dependent oligomerization, but have modified the main text as follows:

*"As shown in **Fig. 1d**, between a concentration of 1 to 3 mM cardiolipin we indeed see a sharp increase in the stabilization of NhaA. From the FSEC traces it is apparent that the increase in stabilization of NhaA from 1 to 3 mM cardiolipin is directly correlated with an increase in the population of NhaA dimers, **Fig. 1e**". Pg 6, line 131.*

Because we no longer refer to the averaged K_d from the separate titrations we have combined all the data into a single curve and refer to the midpoint of the curve, Pg 78, line 774.

Reviewer #2 (Remarks to the Author):

I carefully read the revised manuscript. I think, basically the authors addressed most of my concerns, but I still have two minor concerns as follows before publication of this manuscript.

1. Last time I requested the authors to show a successful application of GFP-TS, which leads to the new membrane protein structure determination or identification of a novel lipid-membrane protein interaction with physiological relevance, not just the validating of the method. While the authors showed the LCP crystallization trials of NhaA in the revised manuscript, I do not think it fully meets my previous request. To properly address my concern, I would like the authors to add the following descriptions and data.

a. The detail of the LCP crystallization is not clear. Did the space group and cell dimension change after addition of cardiolipin? Did the authors see cardiolipin in the LCP structure crystallized in the presence of cardiolipin? I understand that the authors do not want to hurt the novelty of the NhaA LCP structure. So, it is not necessary to describe the details of the cardiolipin binding site, but they should answer the above two questions.

Thank you. To clarify, we do not obtain LCP crystals of NhaA in the absence of cardiolipin so we cannot compare a change in cell group dimensions upon the addition of the lipid. However, we think this is strong evidence that stabilization by cardiolipin is required for LCP crystallization.

We indeed see additional non-protein density around the arginine residues where we have expecting to see cardiolipin binding to NhaA. However, we are unable to describe these details in the paper without depositing the structure in the PDB. Because we are investigating further details with the higher resolution structure (such as the ion-binding site) we would prefer to publish this data as part of a follow-up publication.

b. Similarly, I understand that they do not want to show the names of proteins, whose unpublished structures were determined based on GFP-TS. However, they should describe the results without showing the protein names. For instance, they can say “Protein X, without addition of lipid Y identified by GFP-TS, 7Å resolution. With addition of lipid Y, 2.5Å resolution (Diffraction images)”. They can make this kind of a short list and figure without hurting the novelty of their unpublished structures.

Thank you for this suggestion. We have recently solved the structure of a eukaryotic transporter. We used the GFP-TS assay to establish that the lack of stability in LCP was an issue and we could only obtain LCP crystals in the presence of a stabilizing

ligand. We have made a figure to show these details and, as expected, we have density in the pocket of the LCP structure that fits the bound ligand.

Effect of the crystallization lipid monoolein on membrane protein stability of a eukaryotic transporter. **a.** Supernatant fluorescence of purified fusion before heating $T_m + 5^\circ\text{C}$ (non-filled bars) and that remaining after heating and centrifugation (black bars) in the presence of monoolein. **b.** GFP-TS melting curves for a purified eukaryotic transporter in the absence (black; filled-circles) and presence of either monoolein lipid (black; non-filled circles) or stabilizing ligand (cyan; filled circles) or both monoolein and ligand (cyan; half-filled). **c.** Lipidic cubic phase crystals of eukaryotic transporter shown under a UV-microscope that were obtained in monoolein when 1 mM ligand was added to the purified protein solution for 2 h prior to crystallization. **d.** X-ray diffraction image of an LCP crystal of the eukaryotic transporter crystal grown at 20°C and inset showing non-protein electron density consistent with bound ligand.

We will soon begin to put together a manuscript describing the function and structure of this eukaryotic transporter. Again, with all due respect, we think that this data is best described in a follow up publication.

2. Another concern is about the comments from Reviewer #3. Reviewer #3 has a series of concerns, and I basically agreed to the view of Reviewer #3. So, I think for the acceptance of this manuscript, the support from Reviewer #3 would be required. I would support the publication of this article if Reviewer #3 is happy with the current responses to the reviewer #3.

See response to Reviewer 3.

Reviewer #3 (Remarks to the Author):

The revised manuscript is greatly improved over their original submission. It also includes some new and interesting data. This manuscript nicely adds to a number of key papers pointing to the important role of specific lipid-protein interactions.

Minor comments

I agree with Drew that membrane proteins are difficult to express and purify, the term "large cohort" should be removed. I don't see the point of this and, in my opinion, leaving this in detracts from the paper.

Thank you, we have now removed this term.

The improved X-ray diffraction of NapA is a nice addition. Could the authors include a figure in main text highlighting the diffraction and electron density maps? I also left wondering if cardiolipin is resolved in higher resolution crystals?

See response to Reviewer 2

We understand the reviewer's curiosity. We can reveal that, based on the new data, the interfacial detergent and sulfate ions in our published vapour diffusion structure of NhaA (Lee *et al*, J Gen Physiol 2015) can be re-interpreted as cardiolipin. To illustrate this point more clearly we have generated phenix.polder omit map of our published NhaA dimer structure (shown here at 3.5σ) (Acta Crystallogr D Struct Biol. 2017 Feb 1;73(Pt 2):148-157). CDL fits well into this non-protein density, **Fig. 2a**. Indeed, the two opposing positively charged surfaces are unfavorable that are neutralized by the negatively charged cardiolipin, **Fig.2b, c**.

Again, whilst we agree that the molecular make-up of the binding site for cardiolipin is interesting we feel these structural details are best described in a follow-up publication.

Figure 2. The positioning of cardiolipin between monomers of the NhaA dimer structure. a. The NhaA dimer structure is shown in pink and the omit maps in blue mesh at 3.5σ . **b.** Slice through an electrostatic surface of the NhaA dimer with the modelled CDL between the two protomers. **c.** Top view showing the positively charged interaction surfaces between NhaA protomers and the modelled CDL in between them.

Reviewers' comments:

Reviewer #2 (Remarks to the Author):

I carefully read the revised manuscript. The author tried to address my concerns in the cover letter, but prefers not to include the example of the successful application of GFP-TS in the revised manuscript. I still feel that it is important to show the successful application of GFP-TS to strengthen this paper, and thus strongly recommend the author to include it in the revised manuscript, as I request below. If not, this manuscript would be a kind of weak to publish in Nature Communications. I believe showing these applications would not damage the novelty of the following works.

1. "Thank you. To clarify, we do not obtain LCP crystals of NhaA in the absence of cardiolipin so we cannot compare a change in cell group dimensions upon the addition of the lipid. However, we think this is strong evidence that stabilization by cardiolipin is required for LCP crystallization. We indeed see additional non-protein density around the arginine residues where we were expecting to see cardiolipin binding to NhaA. However, we are unable to describe these details in the paper without depositing the structure in the PDB. Because we are investigating further details with the higher resolution structure (such as the ion-binding site) we would prefer to publish this data as part of a follow-up publication."

I agree that the author does not need to show the details of cardiolipin binding to NhaA. To show the importance of GFP-TS, the author can simply point out in the manuscript that they did not obtain LCP crystals of NhaA in the absence of cardiolipin. It would strengthen the importance of GFP-TS.

2. "Thank you for this suggestion. We have recently solved the structure of a eukaryotic transporter. We used the GFP-TS assay to establish that the lack of stability in LCP was an issue and we could only obtain LCP crystals in the presence of a stabilizing ligand. We have made a figure to show these details and, as expected, we have density in the pocket of the LCP structure that fits the bound ligand.

Effect of the crystallization lipid monoolein on membrane protein stability of a eukaryotic transporter. a. Supernatant fluorescence of purified fusion before heating $T_m + 5^\circ\text{C}$ (non-filled bars) and that remaining after heating and centrifugation (black bars) in the presence of monoolein. b. GFP-TS melting curves for a purified eukaryotic transporter in the absence (black; filled-circles) and presence of either monoolein lipid (black; non-filled circles) or stabilizing ligand (cyan; filled circles) or both monoolein and ligand (cyan; half-filled). c. Lipidic cubic phase crystals of eukaryotic transporter shown under a UV-microscope that were obtained in monoolein when 1 mM ligand was added to the purified protein solution for 2 h prior to crystallization. d. X-ray diffraction image of an LCP crystal of the eukaryotic transporter crystal grown at 20°C and inset showing non-protein electron density consistent with bound ligand.

We will soon begin to put together a manuscript describing the function and structure of this eukaryotic transporter. Again, with all due respect, we think that this data is best described in a follow up publication."

This is a great example of GFP-TS application. So, the figure of "Effect of the crystallization lipid monoolein on membrane protein stability of a eukaryotic transporter.", which is attached in the cover letter, should be included in the revised manuscript. Since it does not show the protein name or ligand name, it would not hurt the novelty of another manuscript describing the function and structure of this eukaryotic transporter.

Reviewer #3 (Remarks to the Author):

The revised manuscript has addressed most of the reviewer's concerns.

As I noted in my first review, the manuscript describes a modification of an established method (ref 13), which originally reported application of lipid screening for thermal stability. This point was noted by two reviewers. In addition, the cardiolipin dependent dimerization of NhaA has previously reported by the authors. Whilst I find the approach and data interesting, the current manuscript is more suitable for another journal, such as Analytical Chemistry. However, including the improved crystal structure of NhaA with resolved cardiolipin bound would warrant publication in Nat Comm. Importantly, this view was also noted by reviewer #2.

The authors state that NhaA retains bound cardiolipin throughout purification (line 89). Does the NhaA sample in their studies have cardiolipin bound? If so, the results for the addition of cardiolipin are difficult to interpret. And does this also suggest there is more than one cardiolipin binding site? The authors need to make note of this point in their paper.

An engineered thermal shift screen reveals specific lipid preferences of eukaryotic and prokaryotic membrane proteins

Corresponding author:

David Drew

We thank the referees for their considered evaluation. We have responded, as appropriate, to all queries below.

Reviewers' comments:

Reviewer #2 (Remarks to the Author):

I agree that the author does not need to show the details of cardiolipin binding to NhaA. To show the importance of GFP-TS, the author can simply point out in the manuscript that they did not obtain LCP crystals of NhaA in the absence of cardiolipin. It would strengthen the importance of GFP-TS.

We have made it clear that LCP NhaA crystals are only obtained with cardiolipin. Together with LCP crystallization of the human SLC35A1 homologue from plant (see below), we think strengthens the use of the GFP-TS method for structural studies.

This is a great example of GFP-TS application. So, the figure of “Effect of the crystallization lipid monoolein on membrane protein stability of a eukaryotic transporter. “, which is attached in the cover letter, should be included in the revised manuscript. Since it does not show the protein name or ligand name, it would not hurt the novelty of another manuscript describing the function and structure of this eukaryotic transporter.

Thank you. Indeed, since we are close to submitting the eukaryotic transporter structure for publication we have decided to take the risk and include all the details of the transporter here, which is a previously unknown CMP-Sialic acid transporter.

We also now show how the GFP-TS assay was used to de-orphanize this transporter. Following the simple premise that ligands which show increased (high) thermostabilization in our assay are also potential substrates, we could quickly identify the correct substrates for plant transporter, and verify our findings against the human homologue. Remarkably, the binding affinity measurements of the substrate against both the plant and human CMP-Sialic acid transporters, using the GFP-TS assay with unpurified membrane samples, matched the ITC binding data of the purified proteins. Given the difficulties in a) establishing functional assays for membrane proteins and b) using ITC to measure binding affinities of membrane proteins (in detergent, not membranes) we think this further data adds significant impact. Lastly, we show that this substrate also stabilized the transporter well enough to obtain well-diffracting LCP crystals in the destabilizing monoolein lipid (after trying for close to five years using conventional crystallization screening methods).

Reviewer #3 (Remarks to the Author):

The revised manuscript has addressed most of the reviewer's concerns.

As I noted in my first review, the manuscript describes a modification of an established method (ref 13), which originally reported application of lipid screening for thermal stability. This point was noted by two reviewers. In addition, the cardiolipin dependent dimerization of NhaA has previously reported by the authors. Whilst I find the approach and data interesting, the current manuscript is more suitable for another journal, such as Analytical Chemistry. However, including the improved crystal structure of NhaA with resolved cardiolipin bound would warrant publication in Nat Comm. Importantly, this view was also noted by reviewer #2.

See response to reviewer #2 and below.

The authors state that NhaA retains bound cardiolipin throughout purification (line 89). Does the NhaA sample in their studies have cardiolipin bound? If so, the results for the addition of cardiolipin are difficult to interpret. And does this also suggest there is more than one cardiolipin binding site? The authors need to make note of this point in their paper.

As shown previously by native MS, only a fraction of NhaA retains cardiolipin during purification and forms stable dimers, while the majority appears to be lipid-free monomers (See Gupta et al, Nature 2017, ext. data figure 7a, and Landreh et al, Nat Commun 2017, Fig. 1e, f). We have modified the text to clarify this (pg 4, lines 88 - 93). In line with these studies, we now show **for the first time that the addition of excess cardiolipin shifts the equilibrium towards the dimeric form of NhaA, demonstrating that the FSEC-TS can robustly detect specific lipid interactions and that cardiolipin does indeed stabilize dimerization.** We agree with the reviewer that this does not necessarily imply a single binding site, but since cardiolipin stabilizes the NhaA dimer it is highly indicative that the negatively charged lipid also binds at the interface, which is the only positively charged surface found in NhaA (it is after all a cation transporter). As we have mentioned in the paper this position would be in full agreement with the electron densities of our previously published NhaA structure (Lee et al, J Gen Physiol 2014) as well as the results from extensive MD simulations (Gupta et al, Nature 2017). In all honestly, whilst we think that deeper mechanistic insights into the exact chemistry and stoichiometry of cardiolipin binding to NhaA is interesting, this is such a small part of the paper and these details go well beyond the scope and main focus of this paper.

REVIEWERS' COMMENTS:

Reviewer #2 (Remarks to the Author):

The authors addressed my concerns by adding the data on the CMP-Sialic acid transporter. I highly recommend the revised manuscript for publication.